# Enhanced subglacial discharge from Antarctica during meltwater pulse 1A

Tao Li [1,2,3] ✉, Laura F. Robinson[2,4], Graeme A. MacGilchrist[5,6], Tianyu Chen [3], Joseph A. Stewart [2], Andrea Burke [6], Maoyu Wang[3], Gaojun Li [3], Jun Chen[3] & James W. B. Rae [6]

Subglacial discharge from the Antarctic Ice Sheet (AIS) likely played a crucial role in the loss of the ice sheet and the subsequent rise in sea level during the last deglaciation. However, no direct proxy is currently available to document subglacial discharge from the AIS, which leaves significant gaps in our understanding of the complex interactions between subglacial discharge and ice-sheet stability. Here we present deep-sea coral $^{234}U/^{238}U$ records from the Drake Passage in the Southern Ocean to track subglacial discharge from the AIS. Our findings reveal distinctively higher seawater $^{234}U/^{238}U$ values from 15,400 to 14,000 years ago, corresponding to the period of the highest iceberg-rafted debris flux and the occurrence of the meltwater pulse 1A event. This correlation suggests a causal link between enhanced subglacial discharge, synchronous retreat of the AIS, and the rapid rise in sea levels. The enhanced subglacial discharge and subsequent AIS retreat appear to have been pre-conditioned by a stronger and warmer Circumpolar Deep Water, thus underscoring the critical role of oceanic heat in driving major ice-sheet retreat.

The uranium isotopic ($\delta^{234}U = (^{234}U/^{238}U_{activity\ ratio} - 1) \times 1000$) composition of seawater is a potential tracer for subglacial discharge and thus ice-sheet stability in the past[1,2]. Due to the relatively mobile nature of $^{234}U$ induced by α-recoil effects[3], $^{234}U$ is preferentially released and transported to the ocean via riverine input[4,5], resulting in an enrichment of $^{234}U$ relative to $^{238}U$ in modern seawater ($\delta^{234}U = 146.8‰$)[6]. Within debris-laden basal ice and subglacial sediments, however, recoil rejection of $^{234}U$ is maintained in either basal ice or subglacial waters, thus leading to a $^{234}U$-enriched reservoir beneath the ice sheets[2]. For example, lake waters derived from the melting of glaciers in the McMurdo Dry Valleys, East Antarctica, have been found to exhibit $\delta^{234}U$ values exceeding 4000‰[7]. High $\delta^{234}U$ values of a similar magnitude have also been observed in chemical precipitates formed in subglacial aquatic environments in East Antarctica[2]. These observations collectively suggest that $^{234}U$-enrichment is likely a prevalent characteristic of subglacial water beneath the Antarctic Ice Sheet (AIS).

Given the widespread presence of subglacial lakes beneath the AIS[8], the pool of excess $^{234}U$ is expected to be considerable, and may have significantly impacted local seawater $\delta^{234}U$ if it was released into the Southern Ocean during episodes of AIS retreat during the last deglaciation. A parallel event occurred due to the collapse of the Laurentide ice sheet during the last deglaciation, resulting in a temporary positive shift in Atlantic $\delta^{234}U$ that overshoots the modern value by 3‰[1]. Nevertheless, the presence of the strong Antarctic Circumpolar Current (ACC) encircling the Antarctic continent implies that any seawater $\delta^{234}U$ anomaly caused by subglacial discharge from the AIS would quickly disperse in the Southern Ocean[9] (Fig. 1). This suggests that reconstructions of Southern Ocean seawater $\delta^{234}U$ are likely to capture such anomalies only during significant subglacial discharge events.

Deep-sea scleractinian corals have been used to reconstruct the evolution of seawater $\delta^{234}U$ in the past[1,10]. The incorporation of U into coral skeletons during their growth not only enables the determination

[1]State Key Laboratory of Palaeobiology and Stratigraphy, Nanjing Institute of Geology and Palaeontology, Chinese Academy of Sciences, Nanjing, China. [2]School of Earth Sciences, University of Bristol, Bristol, UK. [3]Department of Earth and Planetary Sciences, Nanjing University, Nanjing, China. [4] Department of Environment and Geography, University of York, York, UK. [5]Program in Atmospheric and Oceanic Science, Princeton University, Princeton, NJ, USA. [6]School of Earth and Environmental Sciences, University of St Andrews, St Andrews, UK. ✉e-mail: taoli@nigpas.ac.cn

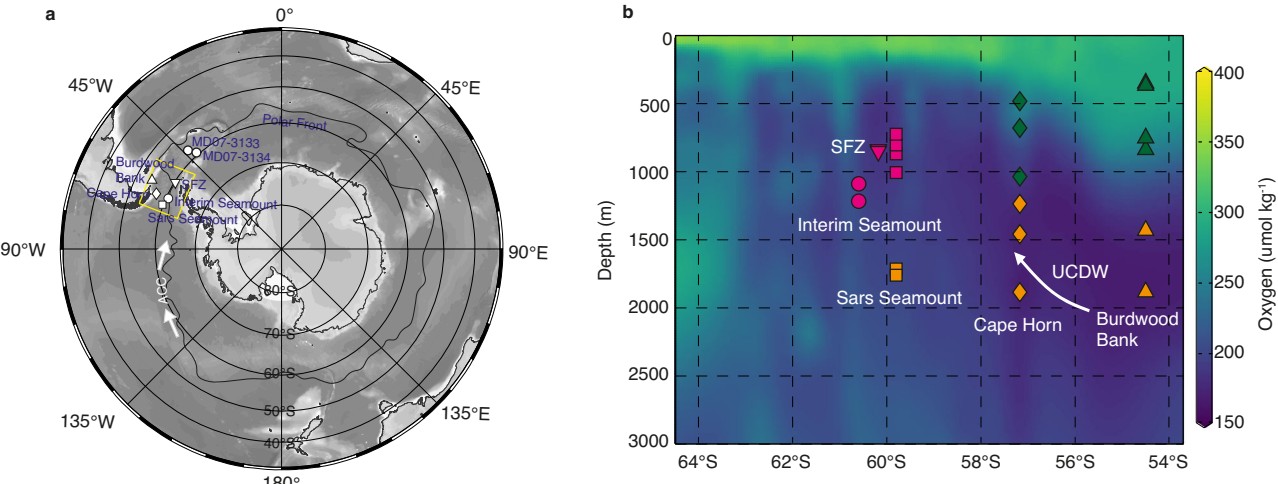

**Fig. 1 | Locations of deep-sea coral samples. a** Map showing the deep-sea coral sites in the Drake Passage, sediment core locations, and the current positions of the polar front (PF)[74]. The white arrows denote the Antarctic Circumpolar Current (ACC). **b** Hydrographic section (yellow rectangle in **a**) showing oxygen content across the Drake Passage. Oxygen content data are from GLODAPv2.2021[75]. Symbols are deep-sea coral sites in this study, with colors denoting different groups of deep-sea corals: Group I (purple), Group II (orange), and Group III (green). UCDW, Upper Circumpolar Deep Water. SFZ, Shackleton Fracture Zone. Figures are plotted using Ocean Data View[76].

of coral ages through U-Th dating but also allows for the reconstruction of $\delta^{234}U$ of contemporaneous seawater ($\delta^{234}U_i$)[11]. However, the reliability of this method faces two primary challenges linked to the preservation of fossil coral samples. Firstly, potential post-mortem diagenetic alteration of the coral skeleton can lead to open-system behavior within U-series isotopes, resulting in biased dates and $\delta^{234}U_i$ values[12]. One common approach to identify altered fossil deep-sea coral samples involves comparing their $\delta^{234}U_i$ with the modern seawater value. This relies on the assumption that seawater $\delta^{234}U$ remains relatively stable over deglacial timescales (-10 thousand years (kyr)) due to the long residence time of U in seawater (~400 kyr)[13,14]. However, a growing body of research suggests that seawater $\delta^{234}U$ is not constant, exhibiting discernible glacial-interglacial cycles[10,15–17], with lower $\delta^{234}U$ during the glacial period than the interglacial period, and resolvable inter-basin difference during the last deglaciation[1]. It is therefore important to deconvolve the relative contributions of diagenetic overprinting versus seawater variations to obtain a reliable Southern Ocean seawater $\delta^{234}U$ record, which requires high sample density within the target age interval. Secondly, U-Th dating of deep-sea corals is complicated by the need to correct for initial $^{230}Th$, introduced by either detrital silicates or Fe-Mn oxides with uncertain $^{230}Th/^{232}Th$ activity ratios. The initial $^{230}Th$ correction, based on measured $^{232}Th$ contents, has a significant impact on the final age results, although its influence on the $\delta^{234}U_i$ values is somewhat muted (Supplementary Fig. 1). Additionally, deep-sea coral samples with high external Th content may potentially undergo open-system processes caused by microbial-driven boring and organic binding of Th[18], further complicating the calculation of $\delta^{234}U_i$ values.

In this study, deep-sea coral samples recovered from seamounts in the Drake Passage, including two sites to the north of the polar front (PF) (Cape Horn and Burdwood Bank), one site aligning with the PF (Sars Seamount), and two sites to the south of the PF (Interim Seamount and Shackleton Fracture Zone (SFZ)), were precisely dated by isotope-dilution U-Th disequilibrium (Fig. 1 and "Methods"). This comprehensive geographic coverage allowed us to investigate a range of oceanographic conditions and potential sources of $\delta^{234}U$ anomalies. Samples were categorized into three groups: (I) shallow Sars Seamount (647–981 m), Interim Seamount (1064–1196 m), and SFZ (806–823 m) that are located in the core region predominantly influenced by eastward ACC are most likely to record any $\delta^{234}U$ anomaly caused by

enhanced subglacial discharge originating from the Antarctic Peninsula and the Amundsen Sea; (II) deep Sars Seamount (1662–1701 m), deep Cape Horn (1214–1877 m), and deep Burdwood Bank (1419–1879 m) that are today bathed by Circumpolar Deep Water (CDW) may help to discern the $\delta^{234}U$ signal originating from other ocean basins; (III) shallow Cape Horn (450–1012 m) and shallow Burdwood Bank (316–894 m) to the north can be used to assess whether the observed $\delta^{234}U$ anomalies are related to discharge from the neighboring South American continent. We analyzed 38 new samples from Sars Seamount 695–981 m, and integrated these data with 335 existing U-Th measurements[19–23] to investigate the presence of $\delta^{234}U$ anomalies in the Southern Ocean at the highest possible temporal resolution. Benefiting from our high sample density, deep-sea coral samples that likely experienced diagenetic alternations were excluded (Supplementary Fig. 2 and "Methods").

## Results and discussion
### Reconstruction of Southern Ocean seawater $\delta^{234}U$
The range of $\delta^{234}U_i$ observed in deep-sea corals of similar ages within a specific location group is noticeably broader than the uncertainties introduced by the analytical process and initial $^{230}Th$ correction (Fig. 2 and Supplementary Fig. 1). This phenomenon holds true across four different deep-sea coral genera (*Desmophyllum*, *Caryophyllia*, *Flabellum*, and *Balanophyllia*), all of which exhibit a comparable degree of variability and follow a common temporal trend (Supplementary Fig. 3). This suggests that species-related effects have a negligible impact on the first-order trend of the $\delta^{234}U_i$ record. Replicate sampling and analysis of six coral samples yielded consistent age results (within uncertainties) but $\delta^{234}U_i$ differences up to 3‰ (Supplementary Fig. 4), which suggests that this degree of variability is likely inherent to deep-sea coral $\delta^{234}U_i$ records[24]. This is further supported by measurements of recent deep-sea corals (<1 thousand years ago (ka)) from different ocean basins, which exhibit a maximum difference of -2.6‰ although their averaged values agree well with modern seawater $\delta^{234}U$ (Supplementary Fig. 5). These interspecimen and intraspecimen $\delta^{234}U_i$ variations are likely attributed to processes such as the internal diffusive movement of uranium[25] and coral vital effects[24], which may obscure the identification of modest $\delta^{234}U_i$ variability. Nevertheless, the large sample density enables us to statistically discern the large-scale trends and noteworthy perturbations, which we delve into in-depth in this study (Fig. 3 and Supplementary Fig. 6).

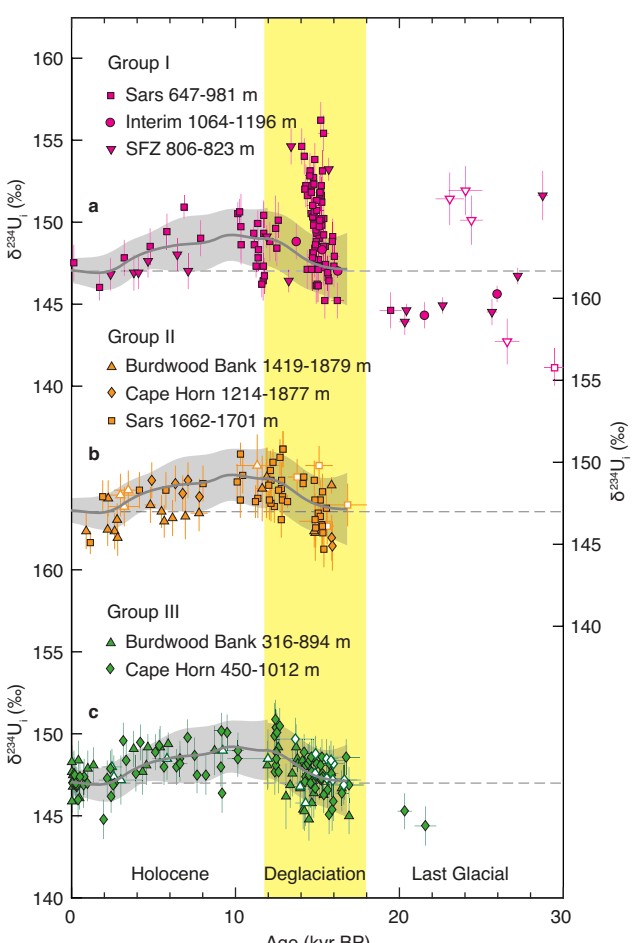

**Fig. 2 | Deep-sea coral δ²³⁴Uᵢ records from the Southern Ocean.** Deep-sea coral δ²³⁴Uᵢ records are splitted into Group I (**a**), Group II (**b**), and Group III (**c**). Symbols are the same as in Fig. 1b. Error bars represent 2σ uncertainties. Open symbols represent samples with a ²³²Th concentration higher than 2 ppb. Gray lines enveloped by shading (±2σ uncertainties) denote the smoothed δ²³⁴U record (excluding samples from Group I) with a 500-yr Gaussian filter. The horizontal dashed lines indicate the δ²³⁴U value of modern seawater[6]. Source data are provided as a Source Data file.

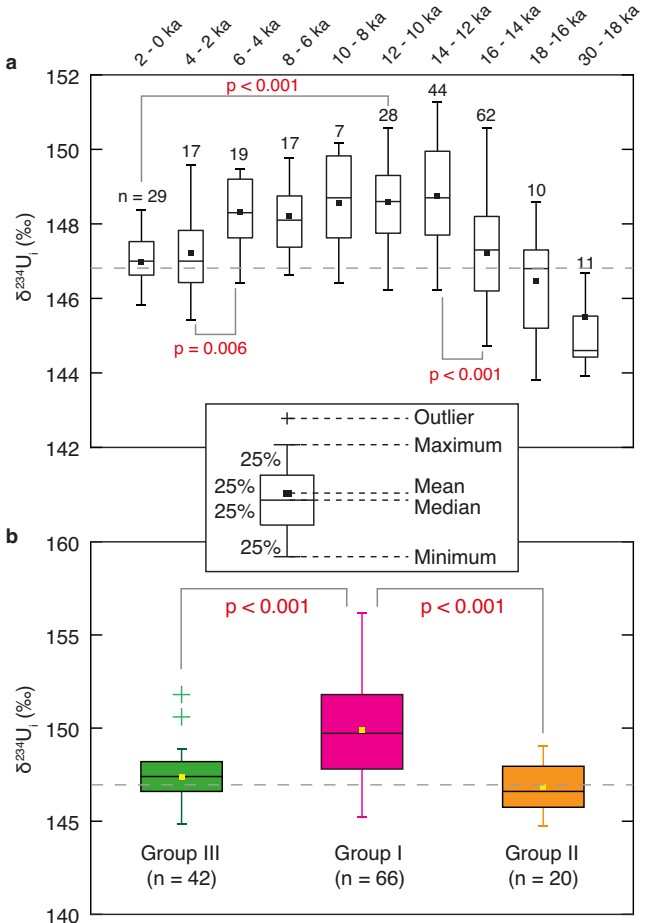

**Fig. 3 | Box plots of deep-sea coral δ²³⁴Uᵢ records. a** Box plots of deep-sea coral δ²³⁴Uᵢ records for each 2-kyr interval over the last 18,000 years and for 30–18 ka (excluding samples from Group I). **b** Box plots of δ²³⁴U results for three groups of deep-sea corals between 16 and 14 ka when high δ²³⁴U values are observed. The *p*-values of a two-sided Wilcoxon rank sum test that assesses the hypothesis of equal δ²³⁴U medians are also shown. Source data are provided as a Source Data file.

Our results reveal a general increase in Southern Ocean seawater δ²³⁴U from the last glacial period to the early Holocene, with a glacial low of ~144.0‰ (~20 ka) to a value of ~148.9‰ during the early Holocene (~10 ka), followed by a gradual decreasing trend towards a modern value of 146.8‰ (i.e., identical to modern open ocean δ²³⁴U[6]) (Fig. 2). When excluding the high δ²³⁴U spikes during the last deglaciation (discussed below), this overall deglacial trend in seawater δ²³⁴U closely aligns with previous studies[1, 10,17,26]. The deglacial increase of seawater δ²³⁴U reflects the input of excess ²³⁴U relative to ²³⁸U to the global ocean, which has been linked to the intensified physical weathering resulting from the rapid retreat of ice sheets during the last deglaciation[1,16]. However, the subsequent Holocene drop of ~2‰ suggests that the decay of ²³⁴U and the gradual mixing with lower-δ²³⁴U water masses has surpassed the input flux of excess ²³⁴U, which is likely related to reduced physical weathering when ice sheets stabilized during the Holocene (Figs. 2 and 3).

### Southern Ocean seawater δ²³⁴U anomaly during the last deglaciation

Superimposed on the general deglacial trend is a remarkable spike in δ²³⁴U reaching up to ~155‰ from ~15.4 to 14 ka (Fig. 2a). Statistical analysis indicates that this δ²³⁴U excursion is restricted to depths of

~1000 m (Group I) within the core region primarily influenced by the eastward ACC, whereas the other sites in the Drake Passage remain unaffected (Fig. 1 and Fig. 3). This high δ²³⁴U signal cannot be fully explained by the advection of ²³⁴U-enriched water from other ocean basins for several reasons. Firstly, a weakened Atlantic Meridional Overturning Circulation during Heinrich Stadial 1 (HS1)[27–29] would imply reduced influence of North Atlantic Deep Water (NADW) on Southern Ocean seawater δ²³⁴U, despite the relatively high δ²³⁴U observed in the North Atlantic[1] (Supplementary Fig. 7). Secondly, the observed δ²³⁴U spike in the Southern Ocean is significantly higher than the peak observed in the North Atlantic, indicating a local source of excess ²³⁴U from the south. Thirdly, deep-sea corals from Group III do not exhibit elevated δ²³⁴U signals (Fig. 2b), which suggests that the excess ²³⁴U is unlikely to be transported from other ocean basins because the Circumpolar Deep Water (CDW) bathing these sites today is a mixture of Atlantic-derived deep waters and recirculating Pacific waters[9]. Moreover, there is also no sign of high δ²³⁴U in the Pacific Ocean during this time interval (Supplementary Fig. 7). The possibility of surface discharge with high δ²³⁴U from South America can also be ruled out since no significant departure from the deglacial δ²³⁴U trend is observed at sites closer to South America (shallow Burdwood Bank and shallow Cape Horn) (Figs. 1 and 3). Therefore, the most plausible explanation for the identified δ²³⁴U spike at depths of ~1000 m near Antarctica is enhanced

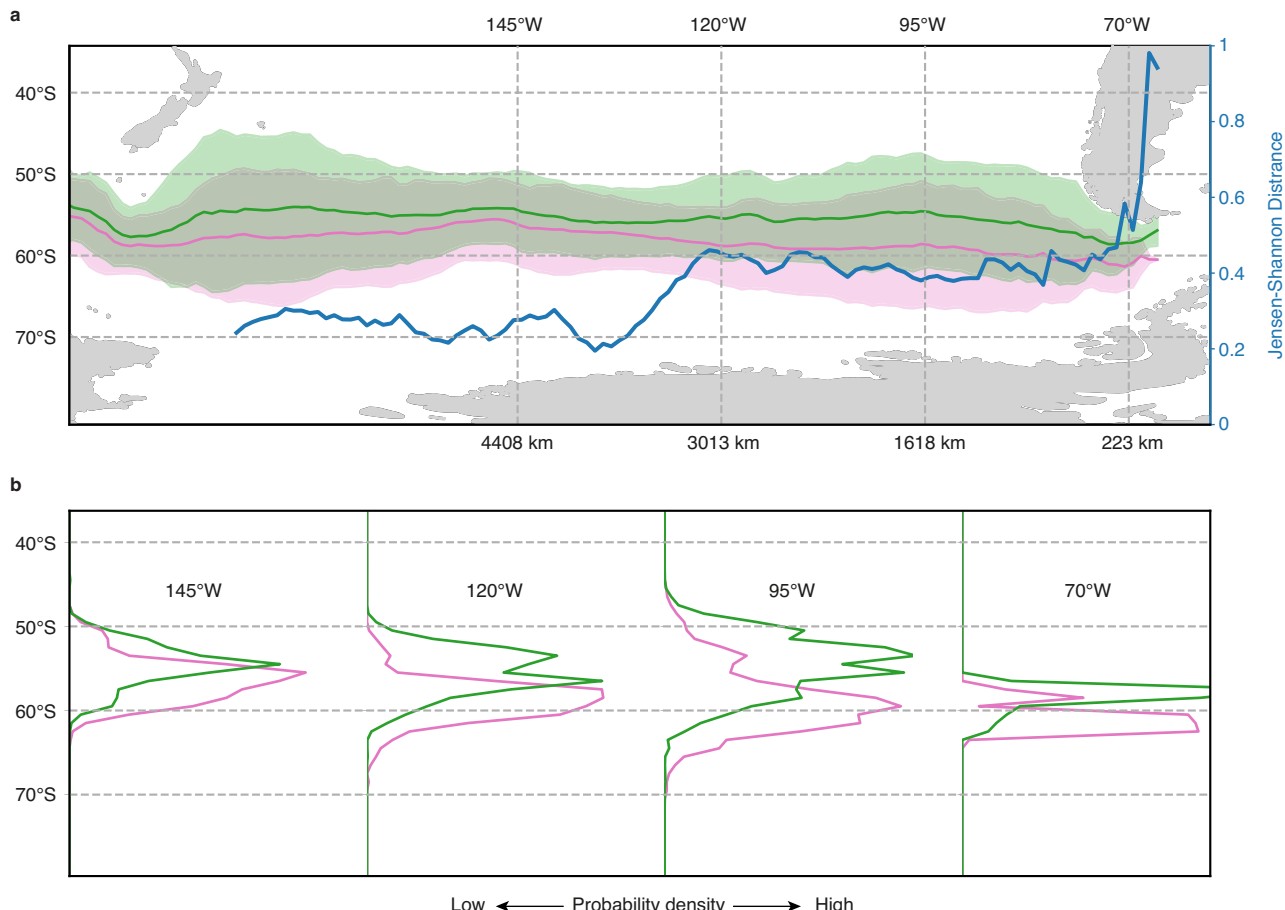

**Fig. 4 | Dispersion of particles initialized in the Drake Passage and traced backwards in time. a** Mean latitude of particles as a function of longitude (solid lines), where the mean direction of motion is East to West, with an envelope corresponding to two standard deviations. The green line denotes particles initialized at the approximate locations and depths corresponding to Group III corals, which do not show the $\delta^{234}$U anomaly. The pink line denotes particles initialized at the approximate locations and depths corresponding to Group I corals, where the $\delta^{234}$U anomaly is evident. The blue line represents the "Jensen–Shannon Distance", a measure of the similarity between the probability density functions (PDFs) of the green and pink particles, where 0 indicates identical distributions and 1 indicates complete independence. **b** PDFs of particle latitude at specific longitudes. The differences in these distributions are most pronounced for localized meltwater sources around the Antarctic Peninsula and are maintained to a degree until -120° W. By contrast, meltwater signals originating further west would become homogenized through mixing, resulting in a similar signal at both sets of coral sites.

subglacial discharge from the AIS during this time period. It is important to note that not all deep-sea coral samples from these sites (Group I) exhibit high $\delta^{234}$U values from -15.4 to 14 ka. Some samples display $\delta^{234}$U$_i$ values within the deglacial trend as constrained by samples from Group II and III. This may reflect the episodic nature of the subglacial discharge events from the AIS, a phenomenon also supported by modern observations[30,31]. Consequently, some short-lived corals may not have captured these transient events.

The geographically and temporally constrained nature of the $\delta^{234}$U spike at the Drake Passage provides additional constraints on the possible source regions of excess $^{234}$U. Specifically, the turbulent nature of ocean circulation means that tracer anomalies tend to be mixed away over rather short distances[32,33]. Consequently, there is a maximum distance that this $\delta^{234}$U anomaly could still be detectable at these specific depths. To visualize this distance in the vicinity of the Drake Passage, we conducted an analysis of backward-in-time trajectories originating from the approximate locations of the deep-sea coral sites in the north and south of the Drake Passage (Fig. 4 and "Methods"). Figure 4a shows the weighted-mean latitude of these trajectories as a function of upstream longitude, bounded by patches spanning two standard deviations both to the north and south. The trajectory latitude serves as a proxy for passive tracer behavior, helping us to understand how far upstream we can expect distinct

properties in the Drake Passage to persist[32]. Our results reveal that, although the weighted-mean latitude of trajectories remains distinct, the distribution of properties becomes increasingly indistinguishable in a relatively short distance upstream (on the order of -3000 km) (Fig. 4). This relatively constant weighted-mean latitude likely reflects the jet-like dynamics of the ACC[34,35]. If the potential source regions of excess $^{234}$U were significantly farther away from the coral sites (e.g., in East Antarctica), the $\delta^{234}$U anomaly would likely be found at both the northern and southern Drake Passage sites. While a more precise determination of the exact origins of excess $^{234}$U would require far tighter constraints on regional circulation during the last deglaciation, we can reasonably conclude that the excess $^{234}$U must have originated from a region within a few thousand kilometers upstream of the Drake Passage, such as the Antarctic Peninsula or the Amundsen Sea (Fig. 4a). Furthermore, it is important to note that vertical mixing away of the $\delta^{234}$U anomaly occurs much more slowly than it does horizontally because the mixing coefficient is three orders of magnitude lower in the vertical than in the horizontal dimension[36]. In other words, the distinct $\delta^{234}$U anomaly observed at -1000 m in the Drake Passage suggests that the excess $^{234}$U may have originated from a similar depth range, which is close to the depth range of the grounding line from which the dense subglacial discharge would emanate[37] (Fig. 1).

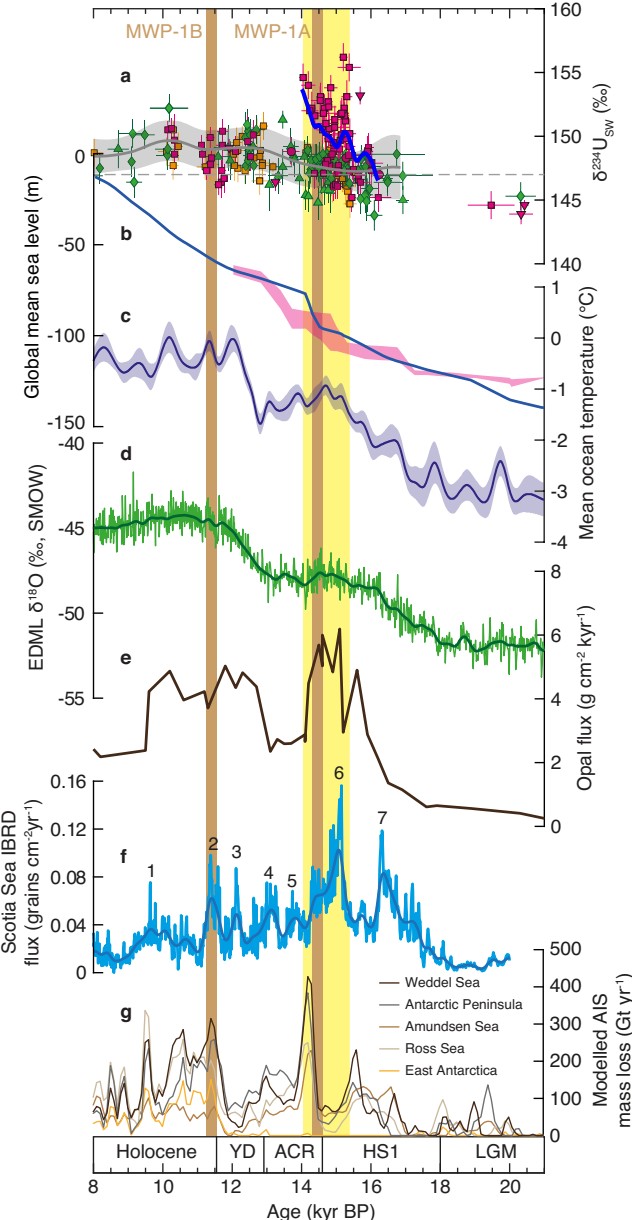

**Fig. 5 | Comparison of Southern Ocean seawater δ²³⁴U record with other paleoclimatic records. a** The seawater δ²³⁴U records reconstructed using deep-sea corals from the Drake Passage of the Southern Ocean. The symbols and the enveloped grey line correspond to those shown in Fig. 2. Error bars represent 2σ uncertainties. The blue line denotes the smoothed record for samples from Group I (Sars Seamount 647–981 m, Interim Seamount 1064–1196 m, and SFZ 806–823 m) with a 50-yr Gaussian filter. **b** The global mean sea-level curve[77] (blue line) and predictions from a glacio-isostatic model[50] (purple shading). **c** Mean ocean temperature as constrained by noble gases trapped in ice cores[60]. The enveloped line shows the splined temperature record with a 2σ uncertainty band. **d** Antarctic ice core δ¹⁸O records (an air temperature proxy) from EPICA Dronning Maud Land (EDML) plotted on the AICC2012 timescale[61,78], and smoothed with a 50-yr Gaussian filter (dark green line). **e** Opal flux in the Atlantic Ocean sector of the Antarctic Zone (core TN057-13-4PC)[62]. **f** Stacked iceberg-rafted debris records from sites MD07-3133 and MD07-3134 in the central Scotia Sea[39]. The numbers denote the Antarctic Ice-sheet Discharge (AID) events. **g** Modeled ice-sheet mass loss history from different sectors of the Antarctic Ice Sheet (AIS) during the last deglaciation[44]. The vertical brown bars denote the two meltwater pulse (MWP) events and the vertical yellow bar highlights the Southern Ocean seawater δ²³⁴U anomaly. ACR, Antarctic Cold Reversal. YD, Younger Dryas. HS1, Heinrich Stadial 1. LGM, Last Glacial Maximum. Source data are provided as a Source Data file.

## Enhanced subglacial discharge and ice-mass loss from the AIS

To investigate the potential temporal and causal connection between enhanced subglacial discharge and the retreat of the AIS during the last deglaciation, we have compared the deep-sea coral δ²³⁴U_i record with published records of iceberg-rafted debris from the Iceberg Alley sites (MD07-3133 and MD07-3134) located ~1000 km downstream from the coral sites in the Southern Ocean[38, 39] (Figs. 1 and 5). Despite the uncertainties associated with the age models and the interpretations of detrital records, their high temporal resolution and proximity to the coral sites allow for a direct comparison with our deep-sea coral δ²³⁴U_i record. It is worth noting that the beryllium (Be) isotope composition of marine sediments serves as a novel proxy for ice-sheet dynamics[40] and has been applied to constrain the retreat history of the AIS[41–43]. However, due to the absence of a continuous Be isotope record offshore of Antarctica that covers the last deglaciation period, it is currently unfeasible to compare the deep-sea coral δ²³⁴U anomaly with a sediment Be isotope record.

Seven AIS discharge events are documented by the iceberg-rafted debris record during the last deglaciation, among which the one at ~15–14 ka (Event 6) stands out with the highest flux of iceberg-rafted debris. Importantly, Event 6 coincides in time with the prominent high δ²³⁴U anomaly recorded by deep-sea corals in the Drake Passage (Fig. 5f). The δ²³⁴U anomaly in the Southern Ocean exhibits a pattern similar to the iceberg-rafted debris record, displaying a relatively abrupt onset at ~15.4 ka and persisting until ~14.0 ka. Although the deep-sea coral sites in the Drake Passage are unlikely to be influenced by downstream subglacial drainages from the Weddell Sea sector of the AIS, as supported by trajectory analysis (Fig. 4), ice-sheet modeling suggests comparable ice-mass loss also from the Antarctic Peninsula and the Amundsen Sea sectors of the AIS during this period[44] (Fig. 5g). Consistent with modern observations of the close linkage between ice-sheet instability and the rapid draining of subglacial water[30, 31,45,46], the fact that the largest AIS discharge event has the same age and duration as the Southern Ocean seawater δ²³⁴U anomaly suggests a causal link between enhanced subglacial discharge and AIS retreat during the last deglaciation.

The Southern Ocean δ²³⁴U anomaly can be further divided into two distinct time intervals (Fig. 5a). From ~15.4 to 14.6 ka, we observe significant fluctuations in δ²³⁴U values ranging from 145.2‰ to 156.2‰. These fluctuations are likely indicative of episodic discharge events from the AIS. Subsequently, from ~14.6 to 14 ka, seawater δ²³⁴U remains consistently elevated, exceeding 151.0‰. This prolonged period of high δ²³⁴U suggests a sustained, catastrophic subglacial discharge from the AIS. Intriguingly, this time interval aligns with the rapid sea-level rise associated with meltwater pulse 1A (MWP-1A, ~14.65 to 14.3 ka) that was initially documented by coral reefs from Barbados in the Caribbean Sea[47,48] and re-constrained by the Tahiti[49] and Great Barrier Reef[50] sea-level records (Fig. 5b). In combination with research on meltwater fingerprinting[49,51,52] and the iceberg-rafted debris record[39], our findings provide evidence supporting an Antarctic contribution to MWP-1A during the last deglaciation. However, because seawater δ²³⁴U anomaly is related to subglacial discharge rather than the freshwater flux resulting from extensive melting, our results do not provide precise constraints on the volume of meltwater originating from the AIS. Consequently, the possible contribution of meltwater from Northern Hemisphere ice sheets, such as Laurentide ice sheet[53–56], Cordilleran ice sheet[53,55,56], and Eurasian ice sheet[57], during MWP-1A cannot be ruled out.

Another meltwater pulse event termed MWP-1B (~11.5 to 11.2 ka) is also documented in the Barbados sea-level records[58]. While both the iceberg-rafted debris record and some model outputs suggest an Antarctic contribution to MWP-1B[39,44], our results do not reveal a significant δ²³⁴U anomaly comparable to that of the MWP-1A (Fig. 5). This is likely associated with reduced subglacial discharge from the AIS,

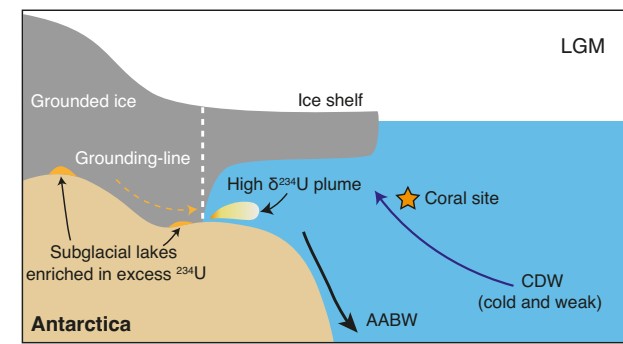

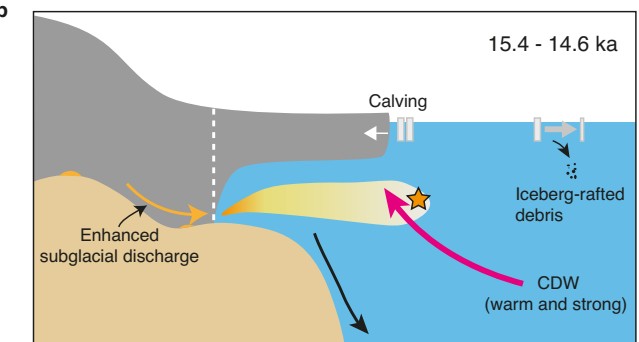

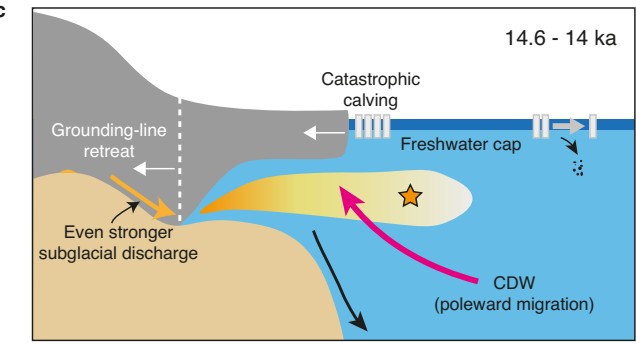

**Fig. 6 | Schematic of changes in ocean circulation, subglacial meltwater plume, and AIS. a** During the Last Glacial Maximum (LGM), the upwelling of warm Circumpolar Deep Water (CDW) and the subglacial meltwater plume were weak. The Antarctic Ice Sheet (AIS) was buttressed by a wide and thick ice shelf. **b** At 15.4 ka, the stronger and warmer CDW triggered the initial stage of AIS retreat by enhancing the basal ice-shelf melting and calving rate. The emergence of enhanced subglacial discharge at this time indicates that the AIS became unstable. The subglacial meltwater plume with abnormally high $\delta^{234}U$ was dispersed and recorded by deep-sea corals at the seamounts close to Antarctica. **c** Freshwater forcing due to basal ice-shelf melting may facilitate poleward migration of CDW that enables the transport of more warm CDW to the base of ice shelves[39], leading to further grounding-line retreat, enhanced ice-sheet calving, and release of more freshwater. More subglacial discharge with high $\delta^{234}U$ and iceberg-rafted debris was released and documented by deep-sea corals and sediment cores, respectively. AABW, Antarctic Bottom Water.

with the high $\delta^{234}U$ signal being rapidly dissipated by the ACC before reaching the coral sites. Further investigation with additional deep-sea coral samples located closer to Antarctica is needed to assess the intensity of the discharge and to test an AIS contribution to sea-level rise during MWP-1B[59].

## Oceanic forcing of AIS retreat

The cause of the enhanced subglacial discharge and subsequent AIS retreat from ~15.4 to 14 ka remains unclear. Here we compare our

deep-sea coral $\delta^{234}U$ record with several climate proxies: the mean ocean temperature record as constrained by noble gases from ice cores[60] (Fig. 5c), an Antarctic ice core $\delta^{18}O$ record[61] (air temperature proxy) (Fig. 5d), and an opal flux record in the Atlantic Ocean sector of the Southern Ocean[62] (Southern Ocean upwelling proxy) (Fig. 5e). The mean ocean temperature record reveals two significant temperature increases of ~1.5 °C, one during the HS1 and the other during the Younger Dryas (YD). The first temperature increase occurs from ~18 to 15.2 ka, immediately preceding the onset of enhanced subglacial discharge from the AIS, as indicated by our deep-sea coral $\delta^{234}U$ record (Fig. 5c). The rapid warming in the subsurface ocean during the HS1 is accompanied by intensified wind-driven upwelling of deep waters in the Southern Ocean[62] (Fig. 5c) and a comparatively milder rise in surface air temperature at Antarctica[61] (Fig. 5d). This suggests that oceanic processes (i.e., a combination of warmer and stronger CDW upwelling) instead of rising surface air temperature, played a pivotal role in conditioning the rapid subglacial discharge and subsequent AIS retreat from ~15.4 to 14 ka. This oceanic thermal forcing may have triggered the initial stage of AIS retreat by enhancing basal ice-shelf melting and calving rates at ~15.4 ka (Fig. 6b). Once initiated, freshwater forcing could have facilitated the poleward migration of warm CDW to the base of ice shelves, as suggested by transient numerical modeling results[39]. This, in turn, would have spurred further grounding-line retreat, ice-sheet calving, and more freshwater release, all contributing to the rapid sea-level rise during MWP-1A (Fig. 6c).

In summary, our high-resolution Southern Ocean seawater $\delta^{234}U$ record provides compelling evidence for enhanced discharge of $^{234}U$-enriched subglacial waters from the AIS during the last deglaciation. This discharge occurred synchronously with the peak in iceberg-rafted debris originating from the Weddell Sea sector of the AIS[39] and MWP-1A[49]. These findings suggest a connection between enhanced subglacial discharge, AIS retreat, and subsequent rapid sea-level rise. We further demonstrate that stronger upwelling of warmer CDW may have preconditioned this rapid AIS retreat and the release of meltwater. Our results therefore underscore the critical role of subsurface warming and enhanced upwelling in driving AIS retreat in the past. Considering the current centennial-scale $CO_2$-driven warming of subsurface waters[63] and southward shifts and strengthening of the westerly winds in the Southern Ocean[64, 65], there is a high risk of AIS retreat by oceanic forcing in the coming centuries.

## Methods

### Sample preparation and U-Th dating

All fossil deep-sea coral samples were collected by research dredge or trawl from the seamounts at the Drake Passage, including Burdwood Bank (54° 30′ S 62° 10′ W) and Cape Horn (57° 10′ S 66° 06′ W) to the north of the Polar Front (PF), Sars Seamount (59° 48′ S 68° 58′ W) and Interim Seamount (60° 36′ S 66° 0′ W) on the PF, and Shackleton Fracture Zone (SFZ, 60° 11′ S 57° 50′ W) to the south of the PF[19–21] (Fig. 1a). Coral samples were collected from water depths ranging from 316 to 1879 m, and from different water masses, including Antarctic Intermediate Water (AAIW), Sub-Antarctic Mode Water (SAMW), Upper Circumpolar Deep Water (UCDW), and Lower Circumpolar Deep Water (LCDW) (Fig. 1b). Samples from Burdwood Bank encompass two major depth ranges, at 334 m and 1879 m, with a relatively high abundance during the Antarctic Cold Reversal (ACR) and the Holocene (Supplementary Fig. 8). The majority of coral samples from Cape Horn were retrieved from a depth of 1012 m and display a high abundance during the HS1, early ACR, early YD, and the Holocene[22]. Sars Seamount coral samples are mainly from 981 m and 1701 m, with samples from 981 m massing at middle to late HS1 and early ACR and samples from 1701 m displaying a high abundance at late HS1 and YD. Samples from SFZ and Interim Seamount cover the last glacial period and are relatively scarce during the last deglaciation and the Holocene. The scarcity of deep-sea corals at great depths in the Drake Passage

during the ACR has been attributed to a northward migration of food supply and poorly oxygenated seawater at depth at that time[22]. Previously, 335 deep-sea coral samples were dated by U-series isotope-dilution techniques[19–23] and here we have added a further 38 samples from Sars Seamount 695–981 m.

The protocol for isotope-dilution U-series dating of the fossil deep-sea corals follows the previously established method[19–21,66]. Approximately 0.3 g of the coral piece was cut and the very surface part of the sample was carefully removed using a Dremel tool. Samples were then extensively cleaned with both oxidizing and reducing chemical procedures[11]. A $^{236}$U-$^{229}$Th mixed spike was added before dissolving the mixture in 2 ml 7 mol/L Optima-grade HNO$_3$. The mixed spike was previously calibrated to the gravimetric standards (388 ppb U solution (CRM-145) and 58 ppt $^{232}$Th solution)[19]. This was followed by co-precipitation of U and Th with pure iron hydroxides before isolating U and Th by passing through anion-exchange columns[67]. U and Th isotopes were measured and corrected by bracketing standard (U112a for U and SGS for Th) on MC-ICPMS (Neptune) at the University of Bristol. A desolvating nebulizer system (Aridus I) was employed to increase the sensitivity of U and Th. A pure $^{236}$U single spike was added to the Th fraction to correct for any drift during the peak jump between $^{229}$Th and $^{230}$Th on the secondary electron multiplier. Mass bias correction was performed using the standard-sample-bracketing (SSB) method. The accuracy and precision were monitored by regularly measuring Harwell uraninite standard (HU1) and ThB standards, which give a long-term external reproducibility of ~1‰ for $^{234}$U/$^{238}$U and ~2‰ for $^{229}$Th/$^{230}$Th, respectively. We use a modern $^{230}$Th/$^{232}$Th ratio of $2 \pm 2 \times 10^{-4}$ (2σ) as measured in unfiltered seawater collected at depths close to the dredge sites for the initial $^{230}$Th correction[68]. Because a relatively large uncertainty in the modern-day $^{230}$Th/$^{232}$Th atomic ratio was applied to correct the initial $^{230}$Th, the final age uncertainties strongly depend on the $^{232}$Th concentrations. All errors associated with the blank correction, mass bias correction, and initial $^{230}$Th correction were propagated with a Monte Carlo method[19–21].

We note that different approaches to calibrate U-series measurements may lead to biased initial δ$^{234}$U values due to the differences between individual aliquots of the secular equilibrium standard (HU1) and between HU1 and the gravimetric standards[16,69]. Since the mixed U-Th spikes used in this study were calibrated to gravimetric standards with older decay constants[11], it becomes necessary to update the decay constant values for both $^{234}$U and $^{230}$Th to make comparisons with more recent dataset[69]. Nevertheless, it should be noted that these corrections for decay constants primarily affect the absolute δ$^{234}$U$_i$ values and do not impact the overall trends observed in the evolution of deep-sea coral δ$^{234}$U$_i$. Given that all deep-sea coral δ$^{234}$U data shown in this study were generated in the same lab and calculated with the same decay constants[19–22], additional decay constant corrections were not performed here. For this reason, the δ$^{234}$U values of modern seawater and recent deep-sea corals reported in this study exhibit a ~2‰ difference when compared to a recent compilation that did account for differences in decay constants and calibration standards[16] (Supplementary Fig. 5).

**Screening criterion**

In this study, fossil deep-sea corals were screened based on analytical uncertainties and their initial δ$^{234}$U values. Firstly, samples with δ$^{234}$U analytical uncertainties exceeding 3‰ were excluded. Although error propagation may introduce some additional uncertainty due to the presence of $^{232}$Th, this influence remains substantially lower than 3‰ (Supplementary Fig. 1). Consequently, δ$^{234}$U analytical uncertainties exceeding 3‰ may be indicative of unstable instrument conditions. 10 out of 373 samples were screened out based on this criterion (Supplementary Dataset S1). We did not reject data based on $^{232}$Th contents because the initial $^{230}$Th correction introduced by contaminants significantly affects the final age results but has a somewhat muted

influence on the δ$^{234}$U$_i$ values (Supplementary Fig. 1). Samples were then screened by δ$^{234}$U$_i$ values using a sliding scale, to account for the fact that Southern Ocean seawater δ$^{234}$U is evolving over the last 30,000 years (Supplementary Fig. 2). We first determined the mean seawater δ$^{234}$U value by smoothing deep-sea coral δ$^{234}$U$_i$ record from Burdwood Bank, Cape Horn, and deep Sars Seamount (1662–701 m). Samples were rejected if their δ$^{234}$U$_i$ values were >3‰ departure from the mean value, considering the upper limits of natural variability of deep-sea coral δ$^{234}$U (Supplementary Fig. 4). It should be noted that the threshold of 3‰ is greater than the observed natural variability (-1.3‰, 2σ) as determined by recent (<1 ka) deep-sea coral from the Southern Ocean (Supplementary Fig. 5). We opted for this threshold to avoid removing samples artificially, recognizing that natural variability might have differed in the past. For samples from Group I (Sars Seamount 647–981 m, Interim Seamount 1064–1196 m, and SFZ 806–823 m), this criterion is not applicable due to the presence of δ$^{234}$U anomaly at the HS1/B-A transition during the last deglaciation. In these cases, the elevated δ$^{234}$U$_i$ values cannot be attributed to open systems, as diagenetic alternations typically result in a positive correlation between $^{234}$U/$^{238}$U and $^{230}$Th/$^{238}$U, a phenomenon often observed in fossil corals from the same terrace[70]. By contrast, the positive shifts of δ$^{234}$U observed here are accompanied by insignificant changes in $^{230}$Th/$^{238}$U (Supplementary Fig. 2), thus supporting the interpretation that these high deep-sea coral δ$^{234}$U$_i$ values capture the rising seawater δ$^{234}$U during the last deglaciation. Apart from the samples from the HS1/B-A transition, two coral samples from Group I with initial δ$^{234}$U values significantly higher than the other samples were omitted from the final dataset. In total, 24 out of 373 samples were removed based on the δ$^{234}$U$_i$ criterion.

**Trajectory analysis**

This analysis uses three-dimensional velocity fields from the Biogeochemical Southern Ocean State Estimate (B-SOSE, iteration 133)[71]. This data-assimilating ocean model approximates the ocean state between 2013 and 2018. The model has a horizontal resolution of 1/6 degrees in both latitude and longitude, allowing it to partially resolve the ocean's turbulent eddy field within the specified region. The velocity field data were saved at 5-day means. Although this is a contemporary ocean model, and the large-scale circulation may have been notably different during the last deglaciation[62], the dynamics of interest for our analysis, i.e., horizontal ocean mixing, are more-than-likely to be consistently represented.

Trajectories were evaluated using the Ocean Parcels python package[72,73]. They are evolved through time using a 4th order Runge–Kutta scheme, with a time-step of 180 min (results presented herein show no sensitivity to this time-step choice). To circumvent complexities arising from advective pathways in the presence of parameterized vertical mixing, trajectories were terminated upon intersecting with the ocean's mixed layer. No additional diffusive parameterization for the trajectories was employed because the turbulent nature of the model circulation already accounts for a significant level of tracer mixing[32].

For the present analysis, trajectories were initialized at the approximate horizontal locations of two of the coral sites. All trajectories were initialized at 66°W; the southern (pink) trajectories were initialized between 59.5° S and 60.5° S, and the northern trajectories were initialized between 56.5° S and 57.5° S. At each site, 8000 trajectories were evenly spaced between latitudes and distributed every 10 m in the depth range of 400–2000 m, during each of the last 12 months of the simulations, for a total of 193,200 trajectories. These were run backward in time for a duration of 6 years, with their position recorded every 30 days. All trajectories were considered together in the analysis.

The Jensen–Shannon distance between probability distributions $P$ and $Q$ is the square-root of the Jensen-Shannon Divergence, which is

given by:

$$\mathrm{JSD}(P\|Q) = \frac{1}{2}\mathrm{KLD}(P\|M) + \frac{1}{2}\mathrm{KLD}(Q\|M) \qquad (1)$$

where $M = (P+Q)/2$ and KLD is the Kullback–Leiber Divergence, a measure of statistical entropy given by:

$$\mathrm{KLD}(P\|M) = \sum \left( P \log \frac{P}{M} \right) \qquad (2)$$

## Data availability

The data generated in this study are presented in the Supplementary Information and are also available on Zenodo https://doi.org/10.5281/zenodo.8433805. Source data are provided with this paper.

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

## Acknowledgements

We thank the participants of cruises LMG06-05, NBP08-05, and NBP11-03 who made this study possible, C. Coath and C. Taylor for their help in the laboratory, and N. Golledge for providing the modeled ice-sheet data. This research is funded by the National Natural Science Foundation of China (41991325, T.L. and T.C.), the Strategic Priority Research Program of Chinese Academy of Sciences (XDB40010200, T.L. and T.C.), the European Research Council, the Natural Environmental Research Council, the U.S. National Science Foundation (PLR-1425989, G.A.M), the UK Research and Innovation (MR/W013835/1, G.A.M), the National Oceanic and Atmospheric Administration (NOAA) Ocean Exploration Trust, and the State Key Laboratory of Palaeobiology and Stratigraphy.

## Author contributions

T.L. and L.F.R. designed the study. L.F.R. and A.B. collected the deep-sea coral samples. T.L., L.F.R., T.C., J.A.S., and A.B. did the U-series analysis. G.A.M. performed the trajectory analysis. T.L., L.F.R., and J.W.B.R. made the interpretations. T.L. and J.W.B.R. wrote the first draft. T.L., L.F.R.,

G.A.M., T.C., J.A.S, A.B., M.W., G.L., J.C., and J.W.B.R. contributed to refinements of the interpretations and editing of the manuscript.

## Competing interests

The authors declare no competing interest.
