## [Peer Review File · Nature Communications]

Enhanced subglacial discharge from Antarctica during meltwater pulse 1AREVIEWER COMMENTS

Reviewer #1 (Remarks to the Author):

(see attached document)

Overview

The manuscript by Li and others reports a study of coral U-series data from the Southern Ocean that identifies a rise in the $\delta^{234}\text{U}$ of seawater during deglaciation. There is a distinct pulse overlapping in time with meltwater pulse 1A, that the authors convincingly relate to recent/nearby discharge from the Antarctic Ice Sheet.

My expertise lies in the realm of U-series geochronology and geochemistry, so much of my feedback focuses on the reporting, screening, and interpretation of this data. While the tracer particle simulations appear robust, I am not able to rigorously evaluate them, and I trust that another reviewer has been able to do so.

Overall, I find the study's results and conclusions compelling and the methods rigorous. However, I would like the authors to more explicitly and quantitatively describe the nuances and complexities of the coral U-series systematics and how they handle them. Clearer and more detailed descriptions of their approach will not only fortify their interpretations but also benefit a broad reading audience of both specialists and non-specialists, alike. To this end, I detail a few general comments, followed by specific line-by-line comments.

I recommend this article for publication in *Nature Communications* so long as the following comments are adequately addressed.

Respectfully,

Graham Edwards

General Comments:

- The authors repeatedly acknowledge that the U-series system is susceptible to minor perturbations (e.g. lines 57–77, 105–14) and dismiss the minor variability in the context of the larger trends of the data. I agree with their conclusions, but I think the manuscript would benefit from a very explicit statement that the study benefits from its dataset's large population, which allows the authors to identify trends within the substantial and significant ($\sim 3\%$) heterogeneity and scatter of the data. The authors should compare the scale of perturbations with the precision of their time- $\delta^{234}\text{U}_i$ record and interpretations thereof. (see comment on line 105–114)
- The manuscript uses abbreviations for sample sites extensively. The clarity of the paper would benefit from a tabulated summary of these. I suggest a legend for Fig. 1 would provide readers with an efficient reference for the abbreviations and corresponding site locations.
- There are a handful of typos and grammatical errors throughout the manuscript. I have made a few suggestions to clarify confusing phrasing, but I encourage the authors to do a close proofreading prior to publication.

Comments by line #:

- 39** The referenced study (Blackburn+ 2020) reports data from subglacial chemical precipitates but not (necessarily) subglacial lake carbonates. While there are U-series data from McMurdo Dry Valley lake carbonates presented in that study, these lake carbonates derive from proglacial, rather than subglacial, lake environments (e.g. Higgins+ 2000, <https://www.jstor.org/stable/521000>). The subglacial precipitates may have formed in subglacial lake environments, but that is not conclusive.
- 73** “has a somewhat muted influence on the $\delta^{234}\text{U}_i$ values” — this assertion and Supplementary Figure 1 should be described in more detail. A more quantitative treatment of this is necessary to establish how insensitive $\delta^{234}\text{U}_i$ is to the ^{230}Th correction. Specifically, I think a more thorough discussion of Supp. Fig. 1 in the methods or in the figure caption (see comment on Supp. Fig. 1 below) would help to elucidate the authors’ point.
- 83–90** This list is very important but the sentence is long and complex. I suggest that authors revise this sentence for clarity and, in particular, reference the specific groupings in Fig. 1 to help readers visualize sample sites.
- 105–14** This is a very important and well-articulated discussion on the inherent variability of coral U-series records. In lines 113–4, the authors state that the influence on large-scale trends is minor. I agree with this statement, but I would like the authors to more quantitatively discuss the relationship between this variability and their results/interpretations. In particular, how does this variability compare to the uncertainty about the trend (i.e. shaded regions in fig. 2). This relates closely to the bulleted comment above: the large dataset allows the authors to resolve a higher signal to noise in the compilation compared to individual samples. Stating this explicitly will strengthen the implied utility of this dataset and the authors’ approach.
- 123–132** This decay-only model overlooks that there is always an input of excess ^{234}U into the ocean (hence the $\delta^{234}\text{U}_i \sim 144\%$ at >20 ka and 146.8% in modern ocean water). The authors should numerically model the $\delta^{234}\text{U}$ evolution with an input flux of ^{234}U that maintains LGM $\delta^{234}\text{U}$ compositions, and examine whether the loss of ^{234}U is still faster than the decline in the $\delta^{234}\text{U}_i$ record. I did a cursory attempt at this calculation: if I impose a flux of ^{234}U sufficient to sustain the system at $\delta^{234}\text{U} = 144\%$ indefinitely and start with a $\delta^{234}\text{U} = 148.9\%$, the $\delta^{234}\text{U}$ of the system decreases by $<1\%$ over 12 ka. To me, this does not suggest that there were additional pulses of high- ^{234}U waters. Rather, the gradual decline of $\delta^{234}\text{U}$ over the Holocene reflects the gradual mixing of ^{234}U -enriched seawater with lower- $\delta^{234}\text{U}$ water masses. I encourage the authors to perform these calculations with their own derivations (lest I made a mistake in mine) to validate these results and adjust their discussion accordingly.
- 136** The authors state “Statistical examination suggests that this $\delta^{234}\text{U}_i$ excursion is restricted to...” with only a reference to Figs. 1 & 2. It is not clear what statistical tests are used and they should be specifically stated.
- 143–5** This sentence is not clear to me. As written, it seems to suggest that the absence of high $\delta^{234}\text{U}$ in the Pacific and Indian oceans during HS1 accounts for the $\delta^{234}\text{U}$ spike in the Southern Ocean. Please either elaborate or rephrase for clarity.

147–9 For clarity, reference Fig. 2C and state that the deep SS water does not show this.

204 While this concordance *suggests* or *indicates* a causal link, I think it does not sufficiently qualify as “direct evidence” for a causal link.

221–2 It seems that the delivery of discharged subglacial water masses to coral-forming locales is somewhat stochastic across the southern ocean based on earlier discussions of water transport. These sampling sites might have just “missed” MWP-1B discharge due to ocean circulation. Just a thought.

Figures

Fig. 1 Provide sample site names for corresponding abbreviations (see bulleted comment above)

Fig. 2 Describe how sample regions are divided among a–c as they relate to Fig. 1.

Fig. 4 (300) The smoothed line in C appears dark green, not black.

Methods

333 Write out “Antarctic Cold Reversal” fully before first use of ACR.

361–4 How is error from ^{230}Th correction calculated? Here the authors state that they use a $^{230}\text{Th}/^{232}\text{Th}$ of 5000. If used as a constant (i.e. with infinite precision), this should contribute no additional uncertainty to the calculation. However, in Supp. Fig 1, they show the effect of ^{232}Th content on “error.” Is there an unstated uncertainty associated with the value of 5000, does the correction end up incorporating uncertainty from an additional isotope measurement that scales with ^{232}Th , or are the authors reporting error as the offset between uncorrected and Th-corrected values? The authors should explicitly state how the described “error” arises.

373–8 While I concur that the internal consistency of the data are not affected by the use of outdated decay constants, the authors should also discuss the effect of these methods on the external reproducibility of ages, since they compare the timeseries with other paleoclimate records (e.g. Fig. 4). While I suspect the differences are minor, this systematic error should be quantified like it is for $\delta^{234}\text{U}_i$. Additionally, per the recommendations of Dutton+ (2017, <https://doi.org/10.1016/j.quageo.2017.03.001>), the authors should state the decay constants used to calibrate standards and calculate ages in the caption or in a footnote to Supplementary Table 1.

387–9 See comment about line 73.

393–4 I believe this 3‰ threshold is derived from the upper limits of variability in Supp. Figs 4 and 5. The authors should reference these to identify the origin of the threshold.

Supplementary Information

Supp. Fig. 1 What do the dashed red lines indicate? Please provide units on the y-axes (years and ‰, presumably?). The authors also need to clarify how they calculated the reported and modeled errors. As in the comments for lines 73 and 361–4, it is not clear how the “error” values reported in this figure were calculated, obscuring confident interpretation of the figure. The authors use “error” and “uncertainty” seemingly interchangeably in the caption, which further challenges interpretation. Without more clarity, this figure fails to adequately substantiate the data screening criteria used in this study.

Supp. Fig. 3 Please describe the black curve and gray band. Presumably this is the filtered timeseries from Figs. 2 and 4, but this should be stated explicitly.

Supp. Fig. 4 Provide information for which samples were measured in duplicate so that readers may cross-reference with the table.

Supp. Fig. 6 See commentary on lines 123–32.

Supp Fig. 7 Provide legend labels for the green and yellow curves or describe in caption.

Reviewer #2 (Remarks to the Author):

Review of the paper by Li et al. entitled as "Enhanced subglacial discharge from Antarctica during meltwater pulse 1A".

Li et al. presented the results of uranium series nuclides measurements on deep sea corals obtained from the Southern Ocean. They found distinctive changes in $\delta^{234}\text{U}$ in seawater during the last deglaciation. The changes are attributed as deglacial climate events since they are likely coincided with other paleoclimate data including sea level changes. In particular, sharp rise in $\delta^{234}\text{U}$ occurred at the time when global sea level rose rapidly known as Mwp1A (melt water pulse 1A). $\delta^{234}\text{U}$ is unique proxy of chemical weathering and authors' data are scientifically sound. Hence, I would like to see the paper be published eventually though I think their discussions need to be modified before accepting it for publication.

The uniqueness of this dataset is that their age model is based on radiometric dates based on uranium series dating. This is very strong point of the data since it is often difficult for paleoceanographic observations obtained from deep sea sediments in the Southern Ocean due to large reservoir effects as well as dilution effects from dead carbon released from Antarctic continent to radiocarbon age models. Hence, I disagree with authors to compare Iceberg rafted debris (IBRD) flux to discuss paleo-environmental conditions because of two reasons. The first is that IBRD has many peaks and their age model is based on correlations to the ice core data under several conditions. Thus, their robustness of the age models for millennial scale changes are very weak. The second, IBRDs are detrital materials remained in deep sea sediments without knowing provenances. No isotope data on IBRD were conducted such Nd and Pb isotopes to identify origins of IBRDs. Therefore, using IBRDs data to discuss Antarctic ice sheet melting history is not convincing. Instead, they should use other studies reconstructing ice sheet history as well as paleoceanography of the Southern Ocean.

Previous Antarctic ice sheet melting history reconstructions include exposure dating using cosmogenic nuclides (eg., Yamane et al., 2010; Johnson et al., 2014; 2020), marine sediment core data using compound specific radiocarbon dating (eg., Johnson et al., 2021; Yamane et al., 2014) and ice core data (eg., Cuffey et al., 2016; Das and Alley, 2008). Further, since $\delta^{234}\text{U}$ is a strong indicator of chemical weathering, authors should consider comparing their results with reactive phase $^{10}\text{Be}/^{9}\text{Be}$ in sediments on and offshore of Antarctica (eg., Sproson et al., 2021, 2022; Behrens et al., 2022; White et al., 2019). Changes in basal condition provide large consequences of ice sheets behavior and are important constraints on ice sheet models. Studies employing $^{10}\text{Be}/^{9}\text{Be}$ in sediments also pointing towards Atmosphere and Ocean interaction related Antarctic ice melting in the past that is in line with currently employed by authors (eg., Yokoyama et al., 2016 SOM; Sproson et al., 2022). Hence, they should discuss their data with above mentioned and beyond data in comprehensive manner to make their data very important.

Because global sea level data from low latitude are reconstructed using U-series dates (Deschamps et al., 2012; Yokoyama et al., 2018), it is useful to expand the section more on comparison between global mean sea level studies. Authors did it for some degrees on Mwp1a, but it would be good to see more focusing on the topic of source(s) of meltwater if it is coming from only Northern hemisphere ice sheets or not (Brendryen et al., 2020). Geophysical fingerprint techniques (eg., Yokoyama and Purcell, 2021) suggested Antarctic involvements (eg., Clark et al., 2002; Deschamps et al., 2012) but far-field fingerprint technique is indirect method so that the current data would provide more constrain on those scenarios combining on authors' previously published dataset (Chen et al., 2016).

Finally, just a minor point but I would not describe U-series dates as absolute dates. Any radiometric dates are based on several assumptions including closed system of nuclides, correctness of decay constant, pristine nature of samples and others. Hence, please change the term to other appropriate one.

Reference cited in this report:

Behrens et al., 2022; *Quat Sci Adv*, 7, 100054
Brendryen et al., 2020; *Nature Geosci.*, 13, 363
Chen et al., 2016; *Science*, 354, 626
Clark et al., 2002; *Nature*, 295, 2438
Cuffey et al., 2016; *PNAS*, 113, 1426
Das and Alley, 2008; *JGR*, 113, D02112
Deschamps et al., 2012; *Nature*, 483, 559
Johnson et al., 2014; *Science*, 343,999
Johnson et al., 2020; *EPSL*, 548, 116501
Johnson et al., 2021; *Nature Geosciences*, 14, 762
Sproson et al., 2021; *Quat Sci Rev*, 256, 106841
Sproson et al., 2022; *Nature Comm*, 13, 2434
White et al., 2019; *EPSL* 505, 86
Yamane et al., 2010; *Jour Quat Sci*, 26, 3
Yamane et al., 2014; *Radiocarbon* 56, 1009
Yokoyama et al., 2016; *PNAS* 113, 2354
Yokoyama et al., 2018; *Nature* 559, 603
Yokoyama and Purcell, 2021; *Geoscience Letters*, 5 article number 1

Review text in black, authors' replies in blue.

Replies to Reviewer #1

The manuscript by Li and others reports a study of coral U-series data from the Southern Ocean that identifies a rise in the $\delta^{234}\text{U}$ of seawater during deglaciation. There is a distinct pulse overlapping in time with meltwater pulse 1A, that the authors convincingly relate to recent/nearby discharge from the Antarctic Ice Sheet.

My expertise lies in the realm of U-series geochronology and geochemistry, so much of my feedback focuses on the reporting, screening, and interpretation of this data. While the tracer particle simulations appear robust, I am not able to rigorously evaluate them, and I trust that another reviewer has been able to do so.

Overall, I find the study's results and conclusions compelling and the methods rigorous. However, I would like the authors to more explicitly and quantitatively describe the nuances and complexities of the coral U-series systematics and how they handle them. Clearer and more detailed descriptions of their approach will not only fortify their interpretations but also benefit a broad reading audience of both specialists and non-specialists, alike. To this end, I detail a few general comments, followed by specific line-by-line comments.

I recommend this article for publication in Nature Communications so long as the following comments are adequately addressed.

Reply: We appreciate the reviewer's positive assessment of our manuscript and concur with the suggestion regarding the need for a more comprehensive description of the coral U-series systematics in the revised manuscript. We have thoroughly reviewed all the comments provided below.

General Comments:

The authors repeatedly acknowledge that the U-series system is susceptible to minor perturbations (e.g. lines 57–77, 105–14) and dismiss the minor variability in the context of the larger trends of the data. I agree with their conclusions, but I think the manuscript would benefit from a very explicit statement that the study benefits from its dataset's large population, which allows the authors to identify trends within the substantial and significant ($\sim 3\%$) heterogeneity and scatter of the data. The authors should compare the scale of perturbations with the precision of their time- $\delta^{234}\text{U}$ record and interpretations thereof. (see comment on line 105–114)

Reply: We appreciate the reviewer's emphasis on the importance of explicitly conveying how our work benefits from the high sample, particularly concerning the interpretation of deep-sea coral $\delta^{234}\text{U}$ data from the Southern Ocean. In response to this concern, we conducted Monte

Carlo simulations and utilized the Wilcoxon rank sum test to examine whether the transient perturbations observed at 16 – 14 ka can be attributed to the internal variability of coral $\delta^{234}\text{U}$ (Supplementary Figure 6).

To perform this analysis, we generated 10,000 synthetic time series of $\delta^{234}\text{U}$ by introducing errors to $\delta^{234}\text{U}$ measurements for different groups of corals (Fig. 3b). These errors were represented as normally distributed random numbers with a mean of zero and specified standard deviations. Subsequently, we calculated p -values, reflecting two-sided Wilcoxon rank sum tests comparing the $\delta^{234}\text{U}$ medians between Group I and Group III, as well as between Group I and Group II, for each time series. Our results unequivocally demonstrate that the perturbations observed in Group I can be statistically distinguished from those in contemporaneous seawater (Group II and III) at 16 – 14 ka, given a standard deviation of 2‰ for coral $\delta^{234}\text{U}$. Considering that the internal $\delta^{234}\text{U}$ variability, as determined by the late Holocene (< 1 ka) samples, is ~1.3‰ (Supplementary Figure 5), it becomes evident that the extensive coral population within the targeted age interval of 16 – 14 ka is more than adequate to differentiate the elevated $\delta^{234}\text{U}$ spikes from those of contemporaneous seawater (Fig. 3). We have thoroughly revised the discussion section and the supplementary information accordingly.

The manuscript uses abbreviations for sample sites extensively. The clarity of the paper would benefit from a tabulated summary of these. I suggest a legend for Fig. 1 would provide readers with an efficient reference for the abbreviations and corresponding site locations.

Reply: Thanks for pointing this out. We have revised the manuscript to reduce the frequency of abbreviations used for the sample sites.

There are a handful of typos and grammatical errors throughout the manuscript. I have made a few suggestions to clarify confusing phrasing, but I encourage the authors to do a close proofreading prior to publication.

Reply: We have carefully checked the words and grammar thoroughly.

Comments by line #:

Line #39 The referenced study (Blackburn+ 2020) reports data from subglacial chemical precipitates but not (necessarily) subglacial lake carbonates. While there are U-series data from Mc-Murdo Dry Valley lake carbonates presented in that study, these lake carbonates derive from proglacial, rather than subglacial, lake environments (e.g. Higgins+ 2000, [https://doi.org/10.1016/S0012-821X\(00\)00000-0](https://doi.org/10.1016/S0012-821X(00)00000-0)).

[//www.jstor.org/stable/521000](http://www.jstor.org/stable/521000)). The subglacial precipitates may have formed in sub-glacial lake environments, but that is not conclusive.

Reply: We agree with the reviewer that the usage of lake carbonates is ambiguous here since subglacial chemical precipitates are not necessarily formed in the lake environment and the form of the chemical precipitates includes both carbonate minerals and amorphous silica/opal precipitation. We have revised this sentence “High $\delta^{234}\text{U}$ values of a similar magnitude have also been observed in chemical precipitates formed in subglacial aquatic environments in East Antarctica”.

Line #73 “has a somewhat muted influence on the $\delta^{234}\text{U}_i$ values” — this assertion and Supplementary Figure 1 should be described in more detail. A more quantitative treatment of this is necessary to establish how insensitive $\delta^{234}\text{U}_i$ is to the ^{230}Th correction. Specifically, I think a more thorough discussion of Supp. Fig. 1 in the methods or in the figure caption (see comment on Supp. Fig. 1 below) would help to elucidate the authors’ point.

Reply: The influence of initial ^{230}Th correction on final age and $\delta^{234}\text{U}_i$ uncertainties were evaluated by calculating the uncertainties of age and $\delta^{234}\text{U}_i$ for a modern sample (age = 0 year, $\delta^{234}\text{U} = 146.8\text{‰}$) and a 20000-year-old sample ($\delta^{234}\text{U}_i = 146.8\text{‰}$) with varying ^{232}Th contents. The modeled uncertainties were calculated by considering a relatively large uncertainty in the initial $^{230}\text{Th}/^{232}\text{Th}$ atomic ratio (2×10^{-4} , 2σ) (Bradt et al., 2009) and were propagated with a Monte Carlo method when solving the age equation (Burke and Robinson, 2012; Chen et al., 2015; Li et al., 2020). We have revised the caption of Supplementary Figure 1 to clarify this.

Lines #83-90 This list is very important but the sentence is long and complex. I suggest that authors revise this sentence for clarity and, in particular, reference the specific groupings in Fig. 1 to help readers visualize sample sites.

Reply: Thanks for pointing out this. We have revised this paragraph as well as Figs. 1 and 2 to clarify this.

Lines #105-114 This is a very important and well-articulated discussion on the inherent variability of coral U-series records. In lines 113–4, the authors state that the influence on large-scale trends is minor. I agree with this statement, but I would like the authors to more quantitatively discuss the relationship between this variability and their results/interpretations. In particular, how does this variability compare to the uncertainty about the trend (i.e. shaded

regions in fig. 2). This relates closely to the bulleted comment above: the large dataset allows the authors to resolve a higher signal to noise in the compilation compared to individual samples. Stating this explicitly will strengthen the implied utility of this dataset and the authors' approach.

Reply: Please see our response to the first general comment.

Lines #123-132 This decay-only model overlooks that there is always an input of excess ^{234}U into the ocean (hence the $\delta^{234}\text{U}_i \sim 144\%$ at >20 ka and 146.8% in modern ocean water). The authors should numerically model the $\delta^{234}\text{U}$ evolution with an input flux of ^{234}U that maintains LGM $\delta^{234}\text{U}$ compositions, and examine whether the loss of ^{234}U is still faster than the decline in the $\delta^{234}\text{U}_i$ record. I did a cursory attempt at this calculation: if I impose a flux of ^{234}U sufficient to sustain the system at $\delta^{234}\text{U} = 144\%$ indefinitely and start with a $\delta^{234}\text{U} = 148.9\%$, the $\delta^{234}\text{U}$ of the system decreases by $<1\%$ over 12 ka. To me, this does not suggest that there were additional pulses of high- ^{234}U waters. Rather, the gradual decline of $\delta^{234}\text{U}$ over the Holocene reflects the gradual mixing of ^{234}U -enriched seawater with lower- $\delta^{234}\text{U}$ water masses. I encourage the authors to perform these calculations with their own derivations (lest I made a mistake in mine) to validate these results and adjust their discussion accordingly.

Reply: We acknowledge the reviewer's valid point that the previous decay-only model employed in our study did not account for the constant riverine input of excess ^{234}U into the ocean. Furthermore, we agree that the gradual decline of $\delta^{234}\text{U}$ observed over the Holocene likely reflects the gradual mixing of ^{234}U -enriched seawater with lower- $\delta^{234}\text{U}$ water masses. To address this, we have added this point into the discussion. As the reviewer suggested, we have checked the calculations with our own simple model, which we elaborate on here. This is a simple calculation that incorporates both ^{234}U decay and the riverine input of excess ^{234}U . We assessed the necessary changes in riverine $\delta^{234}\text{U}$ and riverine U flux relative to the last glacial period to explain the observed 2% drop in seawater $\delta^{234}\text{U}$ over the Holocene (Response figure).

In this model, we assumed that seawater $\delta^{234}\text{U}$ (144‰) was in a steady state at the end of the last glacial period (30 -18 ka). This assumption requires a riverine input of 347‰, taking into account a U residence time of 400 kyr (Dunk et al., 2002; Henderson, 2002). For the last deglaciation (18-11.5 ka), we employed two modeling scenarios: one involving an increase in the $\delta^{234}\text{U}$ of riverine input while keeping the U flux the same and the other involving an increase in U flux while maintaining a constant $\delta^{234}\text{U}$ of 347‰ (red lines). These adjustments were made to align the model outcomes with the deep-sea coral data (grey lines enveloped by

shading). In the context of the Holocene period, we considered two scenarios. Scenario 1 (solid line) demonstrates how changes in seawater $\delta^{234}\text{U}$ can be achieved by decreasing either $\delta^{234}\text{U}$ of riverine input or U flux relative to the last glacial period to align with the deep-sea coral data over the Holocene while scenario 2 (dashed line) reflects the results when either $\delta^{234}\text{U}$ of riverine input or U flux remains consistent with the values of the last glacial period. The model results indicate that either a $\sim 80\%$ drop in $\delta^{234}\text{U}$ of riverine input or a reduction of more than half in riverine U flux compared to the last glacial period is necessary to account for the Holocene decrease in seawater $\delta^{234}\text{U}$ of $\sim 2\%$. Of course a minor combination of both of these factors, as well as mixing with lower $\delta^{234}\text{U}$ waters could also be consistent with the data. Given the under-constrained nature of the calculation and its multiple controls, including uranium mass vs isotopic flux and ocean mixing assumptions, we think a robust exploration of this is beyond the scope of the current study. We have thus removed the decay calculation from the previous figure and opt not to include the figure made for the review response below. We have amended the text as described above in line with the reviewer's important insight on the multiple controls on the Holocene decline in $\delta^{234}\text{U}$ and we hope to explore this further in future work.

Response Figure | The model results depicting relative changes in seawater $\delta^{234}\text{U}$ during the last deglaciation and Holocene periods in comparison to the deep-sea coral $\delta^{234}\text{U}_i$ record from the Southern Ocean. This model assumes that seawater $\delta^{234}\text{U}$ (144‰) was in a steady state at the end of the last glacial period (30 -18 ka). This assumption requires a riverine

input of 347‰, taking into account a U residence time of 400 kyr (Dunk et al., 2002; Henderson, 2002). For the last deglaciation (18-11.5 ka), we employed two modeling scenarios (red lines): one involving an increase in the $\delta^{234}\text{U}$ of riverine input while keeping the U flux the same (**a**) and the other involving an increase in U flux while maintaining a constant $\delta^{234}\text{U}$ of 347‰ (**b**). These adjustments were made to align the model outcomes with the deep-sea coral data (grey lines enveloped by shading). In the context of the Holocene period, we considered two scenarios. Scenario 1 (solid red line) demonstrates how changes in seawater $\delta^{234}\text{U}$ can be achieved by decreasing either $\delta^{234}\text{U}$ of riverine input (**a**) or U flux (**b**) relative to the last glacial period to align with the deep-sea coral data over the Holocene while scenario 2 (dashed red line) reflects the results when either $\delta^{234}\text{U}$ of riverine input (**a**) or U flux (**b**) remains consistent with the values of the last glacial period.

Line #136 The authors state “Statistical examination suggests that this $\delta^{234}\text{U}_i$ excursion is restricted to...” with only a reference to Figs. 1 & 2. It is not clear what statistical tests are used and they should be specifically stated.

Reply: We have divided the previous Fig. 2 into two separate figures: a revised Fig. 2 that exclusively displays the deep-sea coral $\delta^{234}\text{U}_i$ data for different groups, and a new Fig.3, which features box plots for testing the statistical significance of large-scale trends and transient perturbations. In the updated Fig. 3, we employed a two-sided Wilcoxon rank sum test to assess the hypothesis of equal $\delta^{234}\text{U}$ medians between this $\delta^{234}\text{U}_i$ excursion and other $\delta^{234}\text{U}$ data from 16 – 14 ka. This test yielded a p-value less than 10^{-3} , even when considering the internal variability of coral $\delta^{234}\text{U}$ (Fig.3 and Supplementary Figure 6). This statistically significant result demonstrates that the $\delta^{234}\text{U}_i$ excursion (Group I) can be distinguished from the large-scale trends (Group II and III). The main text has been revised to reflect these changes.

Lines #143-145 This sentence is not clear to me. As written, it seems to suggest that the absence of high $\delta^{234}\text{U}$ in the Pacific and Indian oceans during HS1 accounts for the $\delta^{234}\text{U}$ spike in the Southern Ocean. Please either elaborate or rephrase for clarity.

Reply: Thank you for bringing this to our attention. We have revised the sentence and relocated it to the section where we discuss the potential mixing of seawater from other ocean basins.

Lines #147-149 For clarity, reference Fig. 2C and state that the deep SS water does not show this.

Reply: Suggestion accepted. We have revised this sentence to clarify it.

Line #204 While this concordance suggests or indicates a causal link, I think it does not sufficiently qualify as “direct evidence” for a causal link.

Reply: We agree with the reviewer that this concordance does not sufficiently qualify as “direct evidence” and we have revised this sentence: “the fact that the largest AIS discharge event has the same age and duration as the Southern Ocean seawater $\delta^{234}\text{U}$ anomaly suggests a causal link between enhanced subglacial discharge and AIS retreat during the last deglaciation.”

Lines #221-222 It seems that the delivery of discharged subglacial water masses to coral-forming locales is somewhat stochastic across the Southern Ocean based on earlier discussions of water transport. These sampling sites might have just “missed” MWP-1B discharge due to ocean circulation. Just a thought.

Reply: Thanks for the suggestion. We agree that no significant $\delta^{234}\text{U}$ anomaly comparable to that of the MWP-1A is likely associated with reduced subglacial discharge from AIS, with high $\delta^{234}\text{U}$ signal being rapidly dissipated by ACC before reaching the coral sites. Further investigation with additional deep-sea coral samples located closer to Antarctica is needed to assess the intensity of the discharge and to confirm the AIS contribution to sea-level rise during MWP-1B. We have revised the discussion accordingly.

Fig. 1 Provide sample site names for corresponding abbreviations (see bulleted comment above)

Reply: Suggestion accepted. We have revised the manuscript to reduce the frequency of abbreviations used for the sample sites.

Fig. 2 Describe how sample regions are divided among a–c as they relate to Fig. 1.

Reply: We have revised Figs. 1 and 2 as well as figure captions to clearly refer to Fig. 1.

Fig. 4 (300) The smoothed line in C appears dark green, not black.

Reply: Thanks for pointing out this. We have revised the figure caption accordingly.

Line #333 Write out “Antarctic Cold Reversal” fully before first use of ACR.

Reply: Suggestion accepted.

Lines #361-364 How is error from ^{230}Th correction calculated? Here the authors state that they use a $^{230}\text{Th}/^{232}\text{Th}$ of 5000. If used as a constant (i.e. with infinite precision), this should contribute no additional uncertainty to the calculation. However, in Supp. Fig 1, they show the effect of ^{232}Th content on “error.” Is there an unstated uncertainty associated with the value of 5000, does the correction end up incorporating uncertainty from an additional isotope measurement that scales with ^{232}Th , or are the authors reporting error as the offset between uncorrected and Th-corrected values? The authors should explicitly state how the described “error” arises.

Reply: We apologize for the confusion. The $^{230}\text{Th}/^{232}\text{Th}$ ratio should be 2×10^{-4} instead of 5000. The initial ^{230}Th was corrected based on the modern-day $^{230}\text{Th}/^{232}\text{Th}$ atomic ratio of $2 \pm 2 \times 10^{-4}$ (2σ), as determined from unfiltered seawater collected at depths near the dredge sites (Bradtmiller et al., 2009), and the measured ^{232}Th concentrations. Because of the relatively large uncertainty associated with the modern-day $^{230}\text{Th}/^{232}\text{Th}$ atomic ratio, which was propagated with a Monte Carlo method, the final age uncertainties are notably influenced by the ^{232}Th concentrations. We have revised this section to provide clarity.

Lines #373-378 While I concur that the internal consistency of the data are not affected by the use of outdated decay constants, the authors should also discuss the effect of these methods on the external reproducibility of ages, since they compare the timeseries with other paleoclimate records (e.g. Fig. 4). While I suspect the differences are minor, this systematic error should be quantified like it is for $\delta^{234}\text{U}_i$. Additionally, per the recommendations of Dutton+ (2017, <https://doi.org/10.1016/j.quageo.2017.03.001>), the authors should state the decay constants used to calibrate standards and calculate ages in the caption or in a footnote to Supplementary Table 1.

Reply: Although the use of outdated decay constants may bias the calculated $\delta^{234}\text{U}_i$ when compared to the results obtained from other labs (Chutcharavan et al., 2018), it does not affect the absolute age results. This is because the mixed U-Th spikes utilized in this study were calibrated to gravimetric standards using older decay constants (Burke and Robinson, 2012). Given that all deep-sea coral $\delta^{234}\text{U}$ data shown in this study were generated in the same lab and calculated with the same decay constants, additional decay constant corrections were not performed here. We have noted the decay constants used for standard calibration and age calculations in the footnote to Supplementary Table 1, as recommended by the reviewer.

Lines #387-389 See comment about line 73.

Reply: Please see our response to the comments on line #73.

Lines #393-394 I believe this 3 ‰ threshold is derived from the upper limits of variability in Supp. Figs 4 and 5. The authors should reference these to identify the origin of the threshold.

Reply: Suggestion accepted. We have revised the text accordingly.

Supplementary Information

Supp. Fig. 1 What do the dashed red lines indicate? Please provide units on the y-axes (years and ‰, presumably?). The authors also need to clarify how they calculated the reported and modeled errors. As in the comments for lines 73 and 361–4, it is not clear how the “error” values reported in this figure were calculated, obscuring confident interpretation of the figure. The authors use “error” and “uncertainty” seemingly interchangeably in the caption, which further challenges interpretation. Without more clarity, this figure fails to adequately substantiate the data screening criteria used in this study.

Reply: We apologize for any confusion. We have added the units on the y-axis and removed the dashed red lines for clarity. The influence of initial ^{230}Th correction on final age and $\delta^{234}\text{U}_i$ uncertainties were modeled by calculating the uncertainties of age and $\delta^{234}\text{U}_i$ for a modern sample (age = 0 year, $\delta^{234}\text{U} = 146.8\text{‰}$) and a 20000-year-old sample ($\delta^{234}\text{U}_i = 146.8\text{‰}$) with varying ^{232}Th contents and a modern-day $^{230}\text{Th}/^{232}\text{Th}$ atomic ratio of $2 \pm 2 \times 10^{-4}$ (2σ). We have revised the text and the caption of Supplementary Figure 1 to clarify this.

Supp. Fig. 3 Please describe the black curve and gray band. Presumably this is the filtered timeseries from Figs. 2 and 4, but this should be stated explicitly.

Reply: We have revised the caption to clarify this.

Supp. Fig. 4 Provide information for which samples were measured in duplicate so that readers may cross-reference with the table.

Reply: Suggestion accepted. We have added the sample information in the figure so that readers can refer to Supplementary Table 1.

Supp. Fig. 6 See commentary on lines 123–32.

Reply: Please see our response to the comments on lines #123-132.

Supp Fig. 7 Provide legend labels for the green and yellow curves or describe in caption.

Reply: Suggestion accepted.

Replies to Reviewer #2

Li et al. presented the results of uranium series nuclides measurements on deep sea corals obtained from the Southern Ocean. They found distinctive changes in $\delta^{234}\text{U}$ in seawater during the last deglaciation. The changes are attributed as deglacial climate events since they are likely coincided with other paleoclimate data including sea level changes. In particular, sharp rise in $\delta^{234}\text{U}$ occurred at the time when global sea level rose rapidly known as Mwp1A (melt water pulse 1A). $\delta^{234}\text{U}$ is unique proxy of chemical weathering and authors' data are scientifically sound. Hence, I would like to see the paper be published eventually though I think their discussions need to be modified before accepting it for publication.

Reply: We thank the reviewer for the positive comments on our manuscript and we have revised the discussion part as suggested by the reviewer.

The uniqueness of this dataset is that their age model is based on radiometric dates based on uranium series dating. This is very strong point of the data since it is often difficult for paleoceanographic observations obtained from deep sea sediments in the Southern Ocean due to large reservoir effects as well as dilution effects from dead carbon released from Antarctic continent to radiocarbon age models. Hence, I disagree with authors to compare Iceberg rafted debris (IBRD) flux to discuss paleo-environmental conditions because of two reasons. The first is that IBRD has many peaks and their age model is based on correlations to the ice core data under several conditions. Thus, their robustness of the age models for millennial scale changes are very weak. The second, IBRDs are detrital materials remained in deep sea sediments without knowing provenances. No isotope data on IBRD were conducted such Nd and Pb isotopes to identify origins of IBRDs. Therefore, using IBRDs data to discuss Antarctic ice sheet melting history is not convincing. Instead, they should use other studies reconstructing ice sheet history as well as paleoceanography of the Southern Ocean.

Reply: We agree with the reviewer's assessment that the Iceberg rafted debris (IBRD) record is likely hindered by the age model of the sediment core since the dating method relies mainly on the correlation of dust records from these cores to the Antarctic European Project for Ice Coring in Antarctica (EPICA) Dronning Maud Land (EDML) dust record (Weber et al., 2014). Such tuned age models are always based on several assumptions and are typically characterized by a relatively low temporal resolution. While radiocarbon dating of marine carbonates and bulk samples holds the potential to provide robust age constraints, as noted by the reviewer, it often presents challenges for paleoceanographic observations derived from deep-sea sediments in the Southern Ocean. These challenges arise due to significant reservoir effects and the

dilution effects from dead carbon released from the Antarctic continent. Radiocarbon dating applied to sedimentary fatty acids may offer a means to circumvent the influence of dead carbon in sediment (Yamane et al., 2014; Yokoyama et al., 2016). However, its utility may be limited when applied to the last deglaciation and beyond, owing to the concerns about the preservation of organic matter. Furthermore, the calibration of radiocarbon ages to the marine calibration curve is complicated by surface reservoir age, especially for the last deglaciation when surface conditions changed rapidly in the Southern Ocean.

We also agree with the reviewer regarding the uncertainties associated with the sources of IBRDs, which can introduce challenges when utilizing IBRD data to reconstruct the melting history of the Antarctic ice sheet (AIS). As an example, a recent study provided new observations based on sedimentological (grain size and IBRDs) and geochemical records (Nd isotopes) in marine sediment core U1361A (64.41°S, 143.89°E, 3454 m water depth), recovered from the continental rise offshore of the Wilkes Subglacial Basin of EAIS. This study demonstrated that IBRD peaks represent transient events that capture dynamic ice discharge, typically occurring during deglaciation. In contrast, changes in provenance changes sustained for longer durations, suggesting a prolonged shift in the regional glacial erosion locus (Wilson et al., 2018). The absence of Nd isotope data from sediment cores MD07-3133 and MD07-3134 does preclude us from identifying potential variations in IBRD provenance, but our primary focus in this study centers on the dynamic processes of the AIS. Given that the majority of Antarctic icebergs follow a route through Iceberg Alley after calving from the Antarctic margin and traveling counter-clockwise around Antarctica (Stuart and Long, 2011), the IBRD record from Iceberg Alley offers a sensitive and nearly continuous means to reconstruct AIS dynamics by capturing an integrated signal of AIS mass loss (Weber et al., 2014; Weber et al., 2021).

We chose the sediment cores MD07-3133 and MD07-3134 for comparison for two primary reasons. Firstly, their proximity to the coral sites allows us to directly investigate the causal link between enhanced subglacial discharge (as indicated by coral $\delta^{234}\text{U}$) and synchronous AIS retreat (as indicated by IBRDs). Secondly, these cores offer a high temporal resolution, making them particularly valuable for our analysis. According to the original age model developed by Weber et al. (2014), site MD07-3133 exhibits a sedimentation rate of 0.3-2.1 m kyr⁻¹ and MD07-3134 has a sedimentation rate of 0.2-1.2 m kyr⁻¹, which translates into a sample resolution of 5-33 yr and 8-50 yr, respectively. Despite potential uncertainties associated with

age models and provenance changes, the high-resolution IBRD records obtained from MD07-3133 and MD07-3134 represent the most promising candidates for continuous reconstructions of AIS dynamics during the last deglaciation so far.

We have revised the discussion part by mentioning these uncertainties associated with the IBRD record as suggested by the reviewer.

Previous Antarctic ice sheet melting history reconstructions include exposure dating using cosmogenic nuclides (eg., Yamane et al., 2010; Johnson et al., 2014; 2020), marine sediment core data using compound specific radiocarbon dating (eg., Johnson et al., 2021; Yamane et al., 2014) and ice core data (eg., Cuffey et al., 2016; Das and Alley, 2008). Further, since $\delta^{234}\text{U}$ is a strong indicator of chemical weathering, authors should consider comparing their results with reactive phase $^{10}\text{Be}/^9\text{Be}$ in sediments on and offshore of Antarctica (eg., Sproson et al., 2021, 2022; Behrens et al., 2022; White et al., 2019). Changes in basal condition provide large consequences of ice sheets behavior and are important constraints on ice sheet models. Studies employing $^{10}\text{Be}/^9\text{Be}$ in sediments also pointing towards Atmosphere and Ocean interaction related Antarctic ice melting in the past that is in line with currently employed by authors (eg., Yokoyama et al., 2016 SOM; Sproson et al., 2022). Hence, they should discuss their data with above mentioned and beyond data in comprehensive manner to make their data very important.

Reply: We agree with the reviewer regarding the valuable insights provided by reactive phase $^{10}\text{Be}/^9\text{Be}$ records in sediments both on and offshore of Antarctica, offering crucial constraints on past Antarctic ice melting. We are keen to compare our deep-sea coral $\delta^{234}\text{U}$ record with these records as suggested by the reviewer. However, to our best knowledge, the existing marine sediment $^{10}\text{Be}/^9\text{Be}$ dataset in the Southern Ocean is presently limited to the Holocene period (Behrens et al., 2022; Sproson et al., 2021; Sproson et al., 2022; Yokoyama et al., 2016). Regrettably, no continuous reconstructions of sediment $^{10}\text{Be}/^9\text{Be}$ from regions proximal to Antarctica that cover the last deglaciation are available so far. Moreover, even if such a dataset were to emerge, the uncertainties associated with the age model of sediment cores may also pose challenges for the direct comparison with the absolutely dated deep-sea coral $\delta^{234}\text{U}$ record during the last deglaciation.

In response to these considerations, we have revised the discussion section to emphasize the importance of the sediment $^{10}\text{Be}/^9\text{Be}$ record in enhancing our understanding of ice-sheet

dynamics and why it is currently unfeasible to compare the deep-sea coral $\delta^{234}\text{U}$ anomaly with the sediment $^{10}\text{Be}/^9\text{Be}$ record.

Regarding the comparison of deep-sea coral $\delta^{234}\text{U}$ with sediment $^{10}\text{Be}/^9\text{Be}$ record over the Holocene, we have somewhat deemphasized this part because the primary focus of our study is on the remarkable $\delta^{234}\text{U}$ spike during the last deglaciation. The ongoing retreat of the AIS over the Holocene, as constrained by the sediment $^{10}\text{Be}/^9\text{Be}$ record (Yokoyama et al., 2016), may indeed leave discernible imprints in seawater $\delta^{234}\text{U}$ in the Southern Ocean. As we outlined in the introduction section, only a sizeable subglacial discharge event (such as MWP-1A) is likely to be captured by reconstructions of Southern Ocean seawater $\delta^{234}\text{U}$ due to the quick dissipation of excess ^{234}U by the strong Antarctic Circumpolar Current (ACC) that encircles the Antarctic continent. There are no significant perturbations in deep-sea coral $\delta^{234}\text{U}$ record during the Holocene, but this does not necessarily mean no subglacial discharge from AIS. We hope to collect more deep-sea coral samples offshore of Antarctica to compare with the sediment $^{10}\text{Be}/^9\text{Be}$ record in the future to test the link between subglacial discharge and AIS retreat during the Holocene in greater detail.

Because global sea level data from low latitude are reconstructed using U-series dates (Deschamps et al., 2012; Yokoyama et al., 2018), it is useful to expand the section more on comparison between global mean sea level studies. Authors did it for some degrees on Mwp1a, but it would be good to see more focusing on the topic of source(s) of meltwater if it is coming from only Northern hemisphere ice sheets or not (Brendryen et al., 2020). Geophysical fingerprint techniques (eg., Yokoyama and Purcell, 2021) suggested Antarctic involvements (eg., Clark et al., 2002; Deschamps et al., 2012) but far-field fingerprint technique is indirect method so that the current data would provide more constrain on those scenarios combining on authors' previously published dataset (Chen et al., 2016).

Reply: Thanks for the suggestion. We have included the global mean sea level curve as well as GIA model predictions in Fig. 5. Because seawater $\delta^{234}\text{U}$ anomaly in the Southern Ocean is related to subglacial discharge instead of freshwater flux due to intense melting, it is impossible to constrain the amount of meltwater from AIS based on our deep-sea coral $\delta^{234}\text{U}_i$ records. Therefore, our results do not preclude the possible contribution of meltwater from Northern Hemisphere ice sheets during the MWP-1A. We have revised the discussion accordingly.

Finally, just a minor point but I would not describe U-series dates as absolute dates. Any radiometric dates are based on several assumptions including closed system of nuclides, correctness of decay constant, pristine nature of samples and others. Hence, please change the term to other appropriate one.

Reply: Thanks for pointing this out. We have revised the term accordingly.

References:

- Behrens, B.C., Yokoyama, Y., Miyairi, Y., Sproson, A.D., Yamane, M., Jimenez-Espejo, F.J., McKay, R.M., Johnson, K.M., Escutia, C., Dunbar, R.B., 2022. Beryllium isotope variations recorded in the Adélie Basin, East Antarctica reflect Holocene changes in ice dynamics, productivity, and scavenging efficiency. *Quaternary Science Advances* 7.
- Bradtmiller, L.I., Robinson, L.F., McManus, J.F., Auro, M.E., Bostock, H.C., 2009. The distribution of ^{231}Pa and ^{230}Th in paired water column and surface sediment samples. *Geochimica et Cosmochimica Acta Supplement* 73, A154.
- Burke, A., Robinson, L.F., 2012. The Southern Ocean's Role in Carbon Exchange During the Last Deglaciation. *Science* 335, 557-561.
- Chen, T., Robinson, L.F., Burke, A., Southon, J., Spooner, P., Morris, P.J., Ng, H.C., 2015. Synchronous centennial abrupt events in the ocean and atmosphere during the last deglaciation. *Science* 349, 1537-1541.
- Chutcharavan, P.M., Dutton, A., Ellwood, M.J., 2018. Seawater $^{234}\text{U}/^{238}\text{U}$ recorded by modern and fossil corals. *Geochimica et Cosmochimica Acta* 224, 1-17.
- Dunk, R., Mills, R., Jenkins, W., 2002. A reevaluation of the oceanic uranium budget for the Holocene. *Chemical Geology* 190, 45-67.
- Henderson, G.M., 2002. Seawater ($^{234}\text{U}/^{238}\text{U}$) during the last 800 thousand years. *Earth and Planetary Science Letters* 199, 97-110.
- Li, T., Robinson, L.F., Chen, T., Wang, X.T., Burke, A., Rae, J.W.B., Pegrum-Haram, A., Knowles, T.D.J., Li, G., Chen, J., Ng, H.C., Prokopenko, M., Rowland, G.H., Samperiz, A., Stewart, J.A., Southon, J., Spooner, P.T., 2020. Rapid shifts in circulation and biogeochemistry of the Southern Ocean during deglacial carbon cycle events. *Science Advances* 6, eabb3807.
- Sproson, A.D., Takano, Y., Miyairi, Y., Aze, T., Matsuzaki, H., Ohkouchi, N., Yokoyama, Y., 2021. Beryllium isotopes in sediments from Lake Maruwan Oike and Lake Skallen, East Antarctica, reveal substantial glacial discharge during the late Holocene. *Quaternary Science Reviews* 256.
- Sproson, A.D., Yokoyama, Y., Miyairi, Y., Aze, T., Totten, R.L., 2022. Holocene melting of the West Antarctic Ice Sheet driven by tropical Pacific warming. *Nature communications* 13.
- Stuart, K., Long, D., 2011. Tracking large tabular icebergs using the SeaWinds Ku-band microwave scatterometer. *Deep Sea Research Part II: Topical Studies in Oceanography* 58, 1285-1300.

- Weber, M.E., Clark, P.U., Kuhn, G., Timmermann, A., Spreng, D., Gladstone, R., Zhang, X., Lohmann, G., Menviel, L., Chikamoto, M.O., Friedrich, T., Ohlwein, C., 2014. Millennial-scale variability in Antarctic ice-sheet discharge during the last deglaciation. *Nature* 510, 134-138.
- Weber, M.E., Golledge, N.R., Fogwill, C.J., Turney, C.S.M., Thomas, Z.A., 2021. Decadal-scale onset and termination of Antarctic ice-mass loss during the last deglaciation. *Nature communications* 12.
- Wilson, D.J., Bertram, R.A., Needham, E.F., van de Flierdt, T., Welsh, K.J., McKay, R.M., Mazumder, A., Riesselman, C.R., Jimenez-Espejo, F.J., Escutia, C., 2018. Ice loss from the East Antarctic Ice Sheet during late Pleistocene interglacials. *Nature* 561, 383-386.
- Yamane, M., Yokoyama, Y., Miyairi, Y., Suga, H., Matsuzaki, H., Dunbar, R.B., Ohkouchi, N., 2014. Compound-Specific ^{14}C Dating of IODP Expedition 318 Core U1357A Obtained Off the Wilkes Land Coast, Antarctica. *Radiocarbon* 56, 1009-1017.
- Yokoyama, Y., Anderson, J.B., Yamane, M., Simkins, L.M., Miyairi, Y., Yamazaki, T., Koizumi, M., Suga, H., Kushara, K., Prothro, L., Hasumi, H., Southon, J.R., Ohkouchi, N., 2016. Widespread collapse of the Ross Ice Shelf during the late Holocene. *Proceedings of the National Academy of Sciences* 113, 2354-2359.

REVIEWERS' COMMENTS

Reviewer #1 (Remarks to the Author):

I find the authors' replies and revisions wholly satisfactory. I especially appreciate the authors' thorough and thoughtful reply exploring the recovery of $\delta^{234}\text{U}$ from peak conditions ca. 15 ka. I wish them all the best in their future work on this topic.

I have two very minor comments on the current manuscript:

1. In Figure 3, it appears that group I is excluded from panel A. I think this is an appropriate choice. However, it is not clear from the caption what groups of corals are included in the box plots of panel A (only Groups II and III, I think?). I request that the authors explicitly state what groups are included for clarity.
2. On lines 235-6, the authors state "...our findings provide direct evidence supporting an Antarctic contribution to MWP-1A during the last deglaciation." While I find the evidence very compelling, I think there is still some inference involved and "direct evidence" is an overstatement, and I think the word "direct" should be removed.

With these two very minor alterations I think this manuscript is fit for publication.

Reviewer #2 (Remarks to the Author):

I now see that the authors modified the manuscript according to the comments provided except for the term "absolutely dated". They should revised it according to my original comments. Otherwise, I am happy the manuscript be published in the Nature Comm.

Review text in black, authors' replies in blue.

Replies to Reviewer #1

I find the authors' replies and revisions wholly satisfactory. I especially appreciate the authors' thorough and thoughtful reply exploring the recovery of $\delta^{234}\text{U}$ from peak conditions ca. 15 ka. I wish them all the best in their future work on this topic.

Reply: Thanks a lot for your constructive comments on the earlier version which greatly helped to improve the manuscript and thanks again for your assessment of the revised manuscript. We have responded to the comments point-by-point below.

I have two very minor comments on the current manuscript:

1. In Figure 3, it appears that group I is excluded from panel A. I think this is an appropriate choice. However, it is not clear from the caption what groups of corals are included in the box plots of panel A (only Groups II and III, I think?). I request that the authors explicitly state what groups are included for clarity.

Reply: Thanks for pointing out this. Group I is not included in the box plots of panel A and we have revised the caption of Figure 3 to clarify this.

2. On lines 235-6, the authors state "...our findings provide direct evidence supporting an Antarctic contribution to MWP-1A during the last deglaciation." While I find the evidence very compelling, I think there is still some inference involved and "direct evidence" is an overstatement, and I think the word "direct" should be removed.

Reply: Thanks for the suggestion and we have removed the word "direct" from this sentence.

With these two very minor alterations I think this manuscript is fit for publication.

Replies to Reviewer #2

I now see that the authors modified the manuscript according to the comments provided except for the term "absolutely dated". They should revised it according to my original comments. Otherwise, I am happy the manuscript be published in the Nature Comm.

Reply: Thanks again for your assessment of the revised manuscript and we have revised the term "absolutely dated" accordingly.